# Efficient vision Transformer by Information Bottleneck Inspired Token Merging

## Abstract

Self-attention and transformers have been widely used in deep learning. Recent efforts have been devoted to incorporating transformer blocks into different types of neural architectures, including those with convolutions, leading to various vision transformers for computer vision tasks. In this paper, we propose a novel and compact transformer block, Transformer with Information Bottleneck inspired Token Merging, or IBTM. IBTM performs token merging in a learnable scheme. Our IBTM is compatible with many popular and compact transformer networks, such as MobileViT and EfficientViT, and it reduces the FLOPs and the inference time of the vision transformers while maintaining or even improving the prediction accuracy. In the experiments, we replace all the transformer blocks in popular vision transformers, including MobileViT, EfficientViT, ViT, and Swin, with IBTM blocks, leading to IBTM networks with different backbones. The IBTM is motivated by the reduction of the Information Bottleneck (IB), and a novel and separable variational upper bound for the IB loss is derived. The architecture of mask module in our IBTM blocks which generate the token merging mask is designed to reduce the derived upper bound for the IB loss. Extensive results on image classification and object detection evidence that IBTM renders compact and efficient vision transformers with comparable or much better prediction accuracy than the original vision transformers. The code of IBTM is available at `https://anonymous.4open.science/r/IBTM_Transformers-053B/`.

## 1 Introduction

Building upon the success of Transformer in natural language processing (Vaswani et al., 2017), vision transformers have demonstrated remarkable performance across a wide range of tasks (Yuan et al., 2021; Dosovitskiy et al., 2021b; Liu et al., 2021; Zhu et al., 2021; Liang et al., 2021; Cai et al., 2023). However, the achievements of vision transformers are accompanied with heavy computational costs (Dosovitskiy et al., 2021b; Touvron et al., 2021), making their deployment impractical under resource-limited scenarios. The aforementioned limitations have spurred recent research endeavors aimed at developing efficient vision transformers. In this paper, we study the problem of accelerating vision transformers by token merging.

Token merging is an effective method for reducing the FLOPs and improving the inference speed of vision transformers (Han et al., 2015; Zhou et al., 2020; Sun et al., 2021; Kim et al., 2024; Bonnaerens & Dambre, 2023; Bolya et al., 2023). However, most existing token merging methods (Rao et al., 2021; Bolya et al., 2023; Kim et al., 2024; Bonnaerens & Dambre, 2023) largely sacrifice the prediction accuracy of the original transformer networks for reduced computation costs (Bolya et al., 2023; Kim et al., 2024). These methods (Kim et al., 2024; Bolya et al., 2023) generally focus on identifying and merging similar tokens by averaging their features. However, such merging strategies, which are based solely on feature similarity, can potentially diminish the informative features in the tokens that are critical to the prediction tasks. Therefore, it remains an interesting and important question whether we can perform token merging while preserving a compelling performance of the vision transformers after token merging. To this end, we propose a novel transformer block, Transformer with Information Bottleneck inspired Token Merging, or IBTM, which learns how to merge tokens while exhibiting a compelling generalization capability of the transformer with merged tokens.

**Motivation.** Due to the fact that the FLOPs of a vision transformer largely depend on the number of tokens in all the transformer blocks, the FLOPs of a vision transformer can be significantly reduced by reducing the number of tokens in all the transformer blocks. **Our goal is to merge the output tokens of all the transformer blocks into fewer tokens without largely sacrificing the prediction accuracy of the original vision transformer.** However, directly merging the output tokens, even by carefully designed methods (Kim et al., 2024; Bonnaerens & Dambre, 2023; Bolya et al., 2023), would adversely affect the performance of the model. In this paper, we propose to maintain a compelling prediction accuracy of a vision transformer with token merging by an informative token merging process. In our IBTM block, the original attention output tokens of a transformer block are merged into fewer target tokens, and every target token is an informative weighted average of the original output tokens. All the target tokens, or merged tokens are the final attention output tokens for the IBTM block, which are fed to an MLP to produce the output of the IBTM block as illustrated by Figure 1.

Such a token merging process in IBTM is primarily inspired by the well-known presence of considerable redundancy in the original output tokens of transformer blocks (Rao et al., 2021; Bolya et al., 2023). As different tokens have varying importance in modeling the vision features at a particular transformer block, it is natural to compute an informative aggregation of the original attention output tokens as the final (target) attention output tokens so that more informative and more important tokens contribute more to the merged tokens with a larger weight in the weighted average in the aggregation process. A more detailed introduction on the Information Bottleneck (IB) is deferred to Section A in the appendix.

**Contributions.** The contributions of this paper are presented as follows.

First, we present a novel and compact transformer block termed Transformer with Information Bottleneck inspired Token Merging, or IBTM. Our IBTM block generates an informative token merging mask which reduces the IB loss. The IBTM blocks can be used to replace all the transformer blocks in many popular vision transformers, rendering compact vision transformers with competitive performance. The effectiveness of IBTM is evidenced by replacing all the transformer blocks in popular vision transformers, including MobileViT (Mehta & Rastegari, 2022), EfficientViT (Cai et al., 2023), ViT (Dosovitskiy et al., 2021b), and Swin (Liu et al., 2021), with IBTM blocks, for image classification, object detection and instance segmentation tasks.

Second, we propose an informative token merging process for vision transformers, which can reduce the IB loss. As a first step, we derive a novel and *separable* variational upper bound for the IB loss associated with token merging, which is $I(\tilde{X}(G), X) - I(\tilde{X}(G), Y)$ where $I(\cdot, \cdot)$ denotes mutual information and $G$ is the token merging mask in IBTM. $\tilde{X}(G)$, $X$, and $Y$ denote the random variables representing the input features, the learned features, and the labels. We then view a transformer with multiple IBTM blocks as an iterative process for the reduction of the IB loss by gradient descent, and every IBTM block simulates one-step gradient descent on the variational upper bound for the IB loss. Inspired by this understanding, the token merging mask at the current layer is generated from the token merging mask at the previous layer and the input tokens at the current layer by a learnable mask module, following the formula of gradient descent as in (3) in Section 3.2. As a result, such informative token merging process generated in a network with IBTM blocks enjoys reduced IB loss, which is evidenced in our ablation study in Section 4.2. Due to the separability of the variational upper bound for the IB loss, a neural network with IBTM blocks can be trained in an end-to-end manner with standard SGD.

It is worthwhile to mention that our IBTM models can be either fine-tuned from pre-trained backbones or trained from scratch. As evidenced in Table 1, our IBTM models always outperform the currrent state-of-the-art token merging methods, including the fine-tuning-based method LTMP (Bonnaerens & Dambre, 2023), when fine-tuned for the same number of epochs. We remark that as shown in Table 2 in Section 4.2 and Table 7 in Section E.1 of the appendix, the baseline token merging method, ToMe, and LTMP, can already reduce the IB loss. By replacing all the transformer blocks with our IBTM blocks, the networks with IBTM exhibit even smaller IB loss and enjoy higher classification accuracy and less FLOPs, either trained from scratch or fine-tuned from pre-trained models. Furthermore, as shown in Table 3 in Section B.1 of the appendix, our IBTM models also outperform all the competing token merging methods when trained from scratch. Importantly, extensive experiment results on various computer vision tasks demonstrate the compelling performance of IBTM networks compared to the competing baselines.

This paper is organized as follows. The related works in efficient vision transformers and compression of vision transformers by pruning or token merging are discussed in Section 2. The formulation of IBTM is detailed in Section 3. The effectiveness of IBTM is demonstrated in Section 4 for image classification, object detection and instance segmentation tasks, by replacing all the transformer blocks of various popular vision transformers, including MobileViT (Mehta & Rastegari, 2022), EfficientViT (Cai et al., 2023), ViT (Dosovitskiy et al., 2021b), and Swin (Liu et al., 2021), with IBTM blocks. We conclude the paper in Section 5. We use $[n]$ to denote natural numbers between 1 and $n$ inclusively.

## 2 RELATED WORKS

### 2.1 EFFICIENT VISION TRANSFORMERS

Vision transformer models have recently achieved superior performance on a variety of computer vision applications (Dosovitskiy et al., 2021c; Liu et al., 2021; Carion et al., 2020; Zhu et al., 2021; Liang et al., 2021; Wang et al., 2022a). However, vision transformers often encounter high computational demands due to the quadratic complexity of the point-wise attention and numerous Multi-Layer Perceptron (MLP) layers. To mitigate the challenges of high computational costs, various strategies have been developed (Zhu et al., 2021; Yuan et al., 2021), primarily aimed at refining the network architectures and incorporating sparse mechanisms for efficient computation. These include the integration of convolutions into transformer networks (Mehta & Rastegari, 2022; Cai et al., 2023; Liu et al., 2023), the use of knowledge distillation for training more efficient transformers (Graham et al., 2021; Radosavovic et al., 2020; Gong et al., 2022), and compressing existing vision transformers with methods such as pruning (Chen et al., 2021a; Yu et al., 2022a; Kong et al., 2022a). Techniques for compressing vision transformers generally fall into three categories: (1) Channel Pruning, which targets the elimination of superfluous heads and channels within ViT blocks (Chen et al., 2021a; Chavan et al., 2022; Zheng et al., 2022a); (2) Block Pruning, which involves removing redundant transformer blocks (Yu et al., 2022b;a); (3) Token Pruning and Token Merging, which prune less important tokens and merge similar tokens in the input of transformer blocks (Rao et al., 2021; Kong et al., 2022a; Bolya et al., 2023; Wang et al., 2022b; Wu et al., 2023; Xu et al., 2024; Wei et al., 2023).

In this paper, we focus on learning to merge tokens guided by the information bottleneck theory of deep learning and primarily review existing works on Token Pruning and Merging (Wang et al., 2022b; Rao et al., 2021; Bolya et al., 2023; Bonnaerens & Dambre, 2023; Kim et al., 2024). DynamicViT (Rao et al., 2021) observes that the prediction in vision transformers is only based on a subset of the most informative tokens and proposes a hierarchical token sparsification framework to prune redundant tokens. ToMe (Bolya et al., 2023) proposes a graph-based matching algorithm that combines similar tokens in each vision transformer block of a pre-trained vision transformer. LTMP (Bonnaerens & Dambre, 2023) learns threshold masking modules that dynamically determine which tokens to merge and prune in a unified framework similar to DynamicViT. ToFu (Kim et al., 2024) also combines token pruning and token merging. Instead of average merging similar tokens, ToFu proposes a conventional average merging module to improve the quality of merged tokens.

### 2.2 RELATED WORKS ABOUT INFORMATION BOTTLENECK

Saxe et al. (2019) provides the first in-depth analysis of conventional information bottleneck (IB) theories and deep learning to establish the connection between the nonlinearity of neural networks and the compression phase of training. Building on the theory of IB, (Lai et al., 2021) proposes a probabilistic attention module reducing mutual information between the input and the masked representation while increasing mutual information between the masked representation and the task label. Further exploring the mechanics of IB in deep learning, (Zhou et al., 2022) finds that self-attention mechanisms can be interpreted as iterative steps in optimizing the IB objective, which explains the advantages of self-attention in learning robust representation. Distinct from most existing methods that implicitly incorporate the IB principle, our work adopts a direct and innovative approach. We aim to optimize a novel and separable variational upper bound of the IB loss with a learnable token merging method. The proposed IBTM lead to compelling performance on many popular vision transformer architecture with lower computation cost, benefiting from the learnable token merging mechanism guided by the IB principle.

(a) IBTM block for regular transformers, such as ViT and Swin.

(b) IBTM block for efficient transformers, such as MobileViT and EfficientViT.

Figure 1: Overall framework of Information Bottleneck inspired Token Merging (IBTM)-Transformer block for regular transformer blocks such as ViT and Swin (a), and efficient transformer blocks such as MobileViT and EfficientViT (b).

## 3 FORMULATION

In this section, we first illustrate how to perform token merging using a token merging mask. We then describe how to generate the token merging mask from a learnable mask module in a IBTM block, as well as the training algorithm of a neural network with IBTM blocks. We derive a novel and separable variational upper bound for the IB loss, and the token merging masks are generated to reduce such variational upper bound for the IB loss.

### 3.1 INFORMATION BOTTLENECK INSPIRED TOKEN MERGING

Given the input feature tokens $X \in \mathbb{R}^{N \times D}$ where $N$ is the number of tokens and $D$ is the token dimension, the IBTM block first applies the self-attention module on the input feature tokens by $Z = \text{ATTN}(X) \in \mathbb{R}^{N \times D}$, where $\text{ATTN}(\cdot)$ is the regular QKV self-attention operation (Dosovitskiy et al., 2021a). As illustrated in Figure 1, every IBTM block has a learnable mask module that generates the token merging mask $G^{(\ell)}$ where $\ell$ is the index of the current layer or block. The IBTM block merges the $N$ tokens of $Z$ into $P$ tokens with $P < N$ by multiplying $Z$ with the token merging mask $G^{(\ell)} \in \mathbb{R}^{N \times P}$. We set $P = \lceil r \times N \rceil$, where $r \in (0, 1)$ is termed the compression ratio for IBTM, and a smaller $r$ renders less merged tokens after token merging. The token merging mask $G^{(\ell)}$ of the $\ell$-th transformer block is generated by the token merging mask $G^{(\ell-1)}$ of the previous layer and the feature tokens $Z$, which is motivated by reducing the IB loss and detailed in Section 3.2. The token merging mask $G^{(1)}$ for the first transformer block is generated by applying an existing learnable token merging method, LTMP (Bonnaerens & Dambre, 2023), which generates a binarized token merging mask $M \in [0, 1]^{N \times P}$ using Gumbel-Softmax with $N \times P$ learnable parameters. After obtaining the merging mask $G^{(\ell)}$, the features tokens of $Z$ are merged into $P$ tokens by $\tilde{X}(G^{(\ell)}) = \left(Z^\top G^{(\ell)}\right)^\top \in \mathbb{R}^{P \times D}$, which is then passed to the following MLP layers in the transformer block.

In addition to merging tokens in regular transformer blocks such as ViT (Dosovitskiy et al., 2021a) and Swin (Liu et al., 2021), the IBTM block can also be applied to efficient transformer blocks widely applied in efficient vision transformer architectures such as MobileViT (Mehta & Rastegari, 2022) and EfficientViT (Cai et al., 2023). Regular transformer blocks obtain the output by sequentially applying the attention operation and MLP on the input feature tokens. However, efficient transformer blocks usually contain residual connections following the design of residual connections in Convolutional Neural Networks (CNNs). That is, these blocks maintain the same shapes for the input $X$ and the self-attention output $Z$ and concatenate them to produce the output features of the current transformer block. As a result, we cannot only merge the tokens of $Z$. Instead, our IBTM block merges the tokens of both $X$ and $Z$ so that the number of merged tokens for $X$ and $Z$ have is the same. To this end, we apply the same token merging mask $G^{(\ell)}$ to merge both $X$ and $Z$. As a result, the compressed $X$ and $Z$ are of the same shape after the token merging process and they can still be concatenated, which is illustrated in Figure 1b. In addition, transformer

blocks in the efficient vision transformers are usually accompanied with convolution operations so that they need to maintain the feature tokens in a three-dimensional format $X \in \mathbb{R}^{H \times W \times D}$ as illustrated in Figure 1b. To apply our token merging method on efficient transformer blocks, we set the number of merged tokens after token merging as $P = H' \times W'$, where $r$ is the compression ratio, and $H' = \lceil H \times \sqrt{r} \rceil, W' = \lceil W \times \sqrt{r} \rceil$. Therefore, the merged tokens can still be reshaped into three-dimensional features for later convolution operations.

## 3.2 GENERATING TOKEN MERGING MASK BY REDUCING VARIATIONAL UPPER BOUND FOR THE IB LOSS

We describe how to generate the token merging mask in a IBTM block in this subsection, and the generation of the token merging mask is inspired by reduction of the IB loss. We first introduce the setup where the IB loss can be specified.

Given the training data $\{X_i, y_i\}_{i=1}^n$ where $X_i$ is the $i$-the input training feature and $y_i$ is the corresponding class label. Let $Z_i$ be the the self-attention output tokens of the $X_i$, and $\tilde{X}_i(G) = (Z_i G)^\top$ is the merged tokens with $G$ being the token merging mask. We first specify how to compute the IB loss, $\mathrm{IB}(G) = I(\tilde{X}(G), X) - I(\tilde{X}(G), Y)$ which depends on $G$ and other network parameters, $X$ is a random variable representing the input feature which takes values in $\{X_i\}_{i=1}^n$, $\tilde{X}(G)$ is a random variable representing the merged tokens which takes values in $\left\{\tilde{X}_i(G)\right\}_{i=1}^n$. $Y$ is a random variable representing the class label which takes values in $\{y_i\}_{i=1}^n$. Let $\left\{\tilde{\mathcal{C}}_a\right\}_{a=1}^C$ and $\{\mathcal{C}_b\}_{b=1}^C$ be the cluster centroid for the merged tokens and the input features, respectively, where $C$ is the number of classes and the merged tokens or input features with the same training label form a cluster. We also abbreviate $\tilde{X}(G)$ as $\tilde{X}$ for simplicity of the notations. Then we define the probability that $\tilde{X}$ belongs to cluster $\tilde{\mathcal{C}}_a$ as $\Pr\left[\tilde{X} \in a\right] = \frac{1}{n} \sum_{i=1}^n \tau(\tilde{X}_i, a)$ with $\tau(\tilde{X}_i, a) = \frac{\exp\left(-\|\tilde{X}_i - \tilde{\mathcal{C}}_a\|_2^2\right)}{\sum_{a=1}^A \exp\left(-\|\tilde{X}_i - \tilde{\mathcal{C}}_a\|_2^2\right)}$. Similarly, we define the probability that $X_i$ belongs to cluster $\mathcal{C}_b$ as $\Pr[X \in b] = \frac{1}{n} \sum_{i=1}^n \tau(X_i, b)$.

Moreover, we have the joint probabilities $\Pr\left[\tilde{X} \in a, X \in b\right] = \frac{1}{n} \sum_{i=1}^n \tau(\tilde{X}_i, a)\tau(X_i, b)$ and $\Pr\left[\tilde{X} \in a, Y = y\right] = \frac{1}{n} \sum_{i=1}^n \tau(\tilde{X}_i, a)\mathbb{1}_{\{y_i = y\}}$ where $\mathbb{1}_{\{\}}$ is an indicator function. As a result, we can compute the mutual information $I(\tilde{X}(G), X)$ and $I(\tilde{X}(G), Y)$ by

$$I(\tilde{X}(G), X) = \sum_{a=1}^C \sum_{b=1}^C \Pr\left[\tilde{X}(G) \in a, X \in b\right] \log \frac{\Pr\left[\tilde{X}(G) \in a, X \in b\right]}{\Pr\left[\tilde{X}(G) \in a\right] \Pr[X \in b]},$$

$$I(\tilde{X}(G), Y) = \sum_{a=1}^C \sum_{y=1}^C \Pr\left[\tilde{X}(G) \in a, Y = y\right] \log \frac{\Pr\left[\tilde{X} \in a, Y = y\right]}{\Pr\left[\tilde{X}(G) \in a\right] \Pr[Y = y]},$$

and then compute the IB loss $\mathrm{IB}(G)$. As explained in our motivation, we aim to perform token merging while can reduce the IB loss. However, directly optimizing the IB loss in the standard SGD training is difficult as the IB loss is not separable. Given a variational distribution $Q(\tilde{X} \in a | Y = y)$ for $y, a \in [C]$ computed by (9) in the appendix, the following theorem gives a variational upper bound, $\mathrm{IBU}(G)$, for the IB loss $\mathrm{IB}(G)$. $\mathrm{IBU}(G)$ is separable and thus compatible with SGD training with minibatches. $\mathrm{IBU}(G)$ is also referred to as the IB bound in the sequel.

**Theorem 3.1.**

$$\mathrm{IB}(G) \le \mathrm{IBU}(G) - C_0, \tag{1}$$

where $C_0$ is a constant only depending on the input training features $\{X_i\}_{i=1}^n$, and

$$\mathrm{IBU}(G) := \frac{1}{n} \sum_{i=1}^n \sum_{a=1}^C \sum_{b=1}^C \tau(\tilde{X}_i(G), a)\tau(X_i, b) \log \tau(X_i, b) - \frac{1}{n} \sum_{i=1}^n \sum_{a=1}^C \sum_{y=1}^C \tau(\tilde{X}_i(G), a)\mathbb{1}_{\{y_i = y\}} \log Q(\tilde{X} \in a | Y = y).$$

**Proposition 3.2.** Suppose $\tilde{X}_i(G) = \left(Z_i^\top G\right)^\top \in \mathbb{R}^{P \times D}$ with $Z_i \in \mathbb{R}^{N \times D}$ being the self-attention output tokens for the $i$-th training feature and $G \in \mathbb{R}^{N \times P}$ is the token merging mask where $N$ is the number of tokens, $D$ is the token dimension, $P$ is the number of merged tokens after token merging, and $\tilde{X}_i(G)$ denotes the merged tokens. At step $\ell$ of gradient descent on $\text{IBU}(G)$, we have

$$G^{(\ell)} = G^{(\ell-1)} - \eta \nabla_G \text{IBU}(G^{(\ell-1)})$$

$$= G^{(\ell-1)} - \frac{2\eta}{n} \sum_{i=1}^n \sum_{a=1}^C Z_i \frac{S_{ia}^{(l-1)}}{(\gamma_i^{(l-1)})^2} \left(\gamma_i^{(l-1)} \mathcal{C}_a - \zeta_i^{(\ell-1)}\right) \psi_{i,a}, \quad \ell \geq 2, \qquad (2)$$

where $S_{ia}^{(\ell)} := \exp\left(-\left\|\tilde{X}_i(G^{(\ell)}) - \tilde{\mathcal{C}}_a\right\|_2^2\right)$ for $i \in [n]$ and $a \in [C]$, $\gamma_i^{(\ell)} := \sum_{a=1}^C S_{ia}^{(\ell)}$, $\zeta_i^{(\ell)} := \sum_{a=1}^C S_{ia}^{(\ell)} \mathcal{C}_a$ for $i \in [n]$, $\psi_{i,a} := \sum_{b=1}^C \tau(X_i, b) \log \tau(X_i, b) - \sum_{y=1}^C \mathbb{1}_{\{y_i=y\}} \log Q(\tilde{X} \in a | Y = y)$.

The proofs of Theorem 3.1 and Proposition 3.2 are deferred to Section C of the appendix. Inspired by Proposition 3.2, we can understand a transformer with token merging and multiple transformer blocks as an iterative process which reduces $\text{IBU}(G)$ by gradient descent, where the $\ell$-th transformer block performs one-step gradient descent on $\text{IBU}(G)$ according to (2). The mask module of at the $\ell$-th IBTM block generates the token merging mask $G^{(\ell)}$ from $G^{(\ell-1)}$, the token merging mask of the previous block, through (2). To improve the flexibility of the token merging mask, an MLP is applied on $Z_i$. Moreover, as IBU and $\nabla_G \text{IBU}$ are separable, (2) can be performed on a minibatch $\mathcal{B}_j \subseteq \{1, \ldots, n\}$, which is compatible with minibatch-based training with SGD for a transformer network with IBTM blocks. In practice, the mask module of the $\ell$-th IBTM block generates $G^{(\ell)}$ by

$$\tilde{G}^{(\ell)} = G^{(\ell-1)} - \frac{2\eta}{n} \sum_{i \in \mathcal{B}_j} \sum_{a=1}^C Z_i \frac{S_{ia}^{(l-1)}}{(\gamma_i^{(l-1)})^2} \left(\gamma_i^{(l-1)} \mathcal{C}_a - \zeta_i^{(l-1)}\right) \psi_{i,a}, \qquad (3)$$

$$G^{(\ell)} = \tilde{G}^{(\ell)} \circ M^{(\ell)} \qquad (4)$$

where $M^{(\ell)} \in [0,1]^{N \times P}$ is a binarized token merging mask generated by LTMP (Bonnaerens & Dambre, 2023) for the $\ell$-th IBTM block by applying the Gumbel-Softmax operation on $N \times P$ learnable parameters. The mask $M^{(\ell)}$ in our IBTM serves as a learnable token merging mask module. Since the update formulation in Equation (3) does not incorporate any trainable parameters, the number of trainable parameters of an IBTM block is the same as the number of trainable parameters in a transformer block with LTMP, which is $N \times P$. The token merging masks are dynamically generated w.r.t. different inputs following Equation (3) and Equation (4). As shown in Proposition 3.2, the updating formulation of the token merging mask is based on the features at corresponding layers.

Algorithm 1 describes the training process of a neural network with IBTM blocks using the standard cross-entropy loss for a classification problem. It is remarked that all the MLP layers of the mask modules in all the IBTM blocks, along with other network parameters, are updated with standard SGD. In order to generate the token merging masks for all the IBTM blocks before a new epoch starts, we update the variational distribution $Q^{(t)}$ and the clusters $\left\{\tilde{\mathcal{C}}_a^{(t)}\right\}_{a=1}^C$ at the end of the previous epoch.

## 4 EXPERIMENTAL RESULTS

In this section, IBTM-Transformers are assessed for the image classification task on the ImageNet-1k dataset. The results in Section 4.1 indicate that IBTM outperforms existing state-of-the-art networks while maintaining a more compact architecture. In addition, IBTM is compared with existing methods on token merging and shows better performance with lower computation costs. Furthermore, in Sections B.2 and B.3 of the appendix, we demonstrate that the use of IBTM-MobileViT and IBTM-EfficientViT as feature extraction backbones leads to superior mAP and reduced FLOPs compared to the baseline models for the tasks of object detection and semantic segmentation. In Section 4.2, we perform ablation studies on the effects of IBTM in reducing IB loss and the IB loss and IB bound at different layers of a IBTM network.

---

**Algorithm 1** Training Algorithm of IBTMs

---

1: Initialize the weights of the network by $\mathcal{W} = \mathcal{W}(0)$ through random initialization, set $t_{\text{train}}$ which is the number of training epochs.
2: **for** $t \leftarrow 1$ to $t_{\text{train}}$ **do**
3:    **if** $t < t_{\text{warm}}$ **then**
4:       Perform gradient descent by a standard step of SGD without applying token merging in IBTM transformer blocks.
5:    **else**
6:       Update $\tau(\tilde{X}_i, a)$ for all the clusters $a \in [C]$ and $i \in [n]$.
7:       **for** $j \leftarrow 1$ to $J$ **do**
8:          **Forward step**: generate $\left\{ G^{(\ell)} \right\}$ for all the IBTM blocks by (3) using the minibatch $\mathcal{B}_j$, the updated $\left\{ \tau(\tilde{X}_i, a) \right\}_{i \in \mathcal{B}_j, a \in [C]}$, $\left\{ Q^{(t-1)}(\tilde{X} \in a | Y = y) \right\}_{a \in [C], y \in [C]}$, and $\left\{ \tilde{\mathcal{C}}_a^{(t-1)} \right\}_{a=1}^{C}$, as well as the output of the network
9:          **Backward step**: update the MLP layers of the mask modules in all the IBTM blocks as well as all the other weights in the neural network by a standard step of SGD on the cross-entropy loss
10:       **end for**
11:       Compute $Q^{(t)}(\tilde{X} \in a | Y = y)$ by Eq. (9) in the appendix, and update the cluster centroids $\left\{ \tilde{\mathcal{C}}_a^{(t)} \right\}_{a=1}^{C}$.
12:    **end if**
13: **end for**
14: **return** The trained weights $\mathcal{W}$ of the network

---

## 4.1 IMAGE CLASSIFICATION

**Implementation details.** In this section, we evaluate IBTM models for ImageNet (Russakovsky et al., 2015) classification. We employ MobileViT-S (Mehta & Rastegari, 2022), MobileViT-XS (Mehta & Rastegari, 2022), EfficientViT-B1 (Cai et al., 2023), ViT-S (Dosovitskiy et al., 2021a), ViT-B (Dosovitskiy et al., 2021a), Swin-T (Liu et al., 2021), and Swin-B (Liu et al., 2021) as backbone architectures. We substitute the conventional transformer blocks in these backbones with IBTM blocks. Implementation details on adapting token merging to hierarchical vision transformers such as Swin Transformers are deferred to Section D.2 in the appendix. All the experiments are conducted on four NVIDIA A100 GPUs with a total batch size of 1024 images. Following prior works (Liu et al., 2021), our training incorporates popular data augmentation methods such as RandAugment, Mixup, Cutmix, and random erasing. We set $\eta$ in Equation (2) to 1 in all the experiments. In addition, we apply a softmax operation on the token merging mask at each layer to ensure the aggregation weights for each merged token sum to 1. In all our experiments, we set the value of compression ratio $r = 0.7$ for all our IBTM models. A study on the impact of the compression ratio $r$ to the performance of the IBTM model is performed in Table 8 in Section E.2 of the appendix.

We conduct the experiments of IBTM for the token merging of vision transformers under two different training setups, which are the fine-tuning setup and the training-from-scratch setup. The experiments in the fine-tuning setup are conducted following the state-of-the-art token merging method, LTMP (Bonnaerens & Dambre, 2023). The training-from-scratch setup is designed to explore the potential of training IBTM-Transformers from the beginning while reducing the IB loss with token merging, and the training with different backbones follows the same training settings as the original training process of the corresponding backbones (Mehta & Rastegari, 2022; Cai et al., 2023; Dosovitskiy et al., 2021a; Liu et al., 2021).

### 4.1.1 FINE-TUNING SETUP

Our proposed IBTM can be straightforwardly applied to token merging with pre-trained models using the fine-tuning setup as in the existing state-of-the-art token merging method, LTMP (Bonnaerens & Dambre, 2023). In the fine-tuning setup, IBTM models are not trained from scratch, and token merging for a pre-trained visual transformer can be performed by simply changing all the transformer blocks of the pre-trained models to IBTM-Transformer blocks according to Section 3.1. Following the settings in LTMP (Bonnaerens & Dambre, 2023), the token merging mask modules are added to the original transformer blocks, and all the pre-trained weights are loaded as the ini-

tialization for the IBTM models. In the fine-tuning process, the pre-trained weights are not updated and only the weights in the token merging mask modules, $\{M^{(\ell)}\}$, are updated. We fine-tune the IBTM models for 1, 5, 10, 25, and 50 epochs, respectively, and compare them with LTMP models fine-tuned for the same number of epochs. Note that IBTM models and LTMP models with the same backbones have the same number of parameters.

| Methods | # Params. | FLOPs | Inference Time (ms/batch) | Top-1 Accuracy (%) | | | | | |
|---|---|---|---|---|---|---|---|---|---|
| | | | | 0 | 1 | 5 | 10 | 25 | 50 |
| MobileViT-XS (Mehta & Rastegari, 2022) | 2.3 M | 0.70 G | 11.3 | 74.80 | - | - | - | - | - |
| ToMe-MobileViT-XS (Bolya et al., 2023) | 2.3 M | 0.54 G | 10.4 | 72.73 | - | - | - | - | - |
| ToFu-MobileViT-XS (Kim et al., 2024) | 2.3 M | 0.54 G | 10.7 | 73.32 | - | - | - | - | - |
| LTMP-MobileViT-XS(Bonnaerens & Dambre, 2023) | 2.3 M | 0.56 G | 10.9 | - | 73.91 | 77.69 | 73.98 | 74.05 | 74.18 |
| **IBTM-MobileViT-XS (Fine-tuned)** | 2.3 M | 0.52 G | 10.3 | - | **74.25** | **74.31** | **74.54** | **74.70** | **74.95** |
| MobileViT-S (Mehta & Rastegari, 2022) | 5.6 M | 1.40 G | 15.1 | 78.40 | - | - | - | - | - |
| ToMe-MobileViT-S (Bolya et al., 2023) | 5.6 M | 1.22 G | 14.2 | 76.72 | - | - | - | - | - |
| ToFu-MobileViT-S (Kim et al., 2024) | 5.6 M | 1.22 G | 14.4 | 77.24 | - | - | - | - | - |
| LTMP-MobileViT-S(Bonnaerens & Dambre, 2023) | 5.6 M | 1.26 G | 14.5 | - | 77.53 | 77.69 | 77.82 | 78.03 | 78.14 |
| **IBTM-MobileViT-S (Fine-tuned)** | 5.6 M | 1.17 G | 14.1 | - | **77.72** | **78.15** | **78.34** | **78.85** | **79.05** |
| EfficientViT-B1 (Cai et al., 2023) | 9.1 M | 0.52 G | 10.0 | 79.40 | - | - | - | - | - |
| ToMe-EfficientViT-B1 (Bolya et al., 2023) | 9.1 M | 0.47 G | 9.6 | 78.81 | - | - | - | - | - |
| ToFuEfficientViT-B1 (Kim et al., 2024) | 9.1 M | 0.47 G | 9.8 | 79.04 | - | - | - | - | - |
| LTMP-EfficientViT-B1(Bonnaerens & Dambre, 2023) | 9.1 M | 0.50 G | 9.8 | - | 79.21 | 79.31 | 79.32 | 79.36 | 79.40 |
| **IBTM-EfficientViT-B1 (Fine-tuned)** | 9.1 M | 0.44 G | 9.6 | - | **79.39** | **79.62** | **79.85** | **80.07** | **80.22** |
| Swin-T (Liu et al., 2021) | 29.0 M | 4.50 G | 20.8 | 81.30 | - | - | - | - | - |
| ToMe-Swin-T (Bolya et al., 2023) | 29.0 M | 3.91 G | 17.5 | 79.28 | - | - | - | - | - |
| ToFuSwin-T (Kim et al., 2024) | 29.0 M | 3.91 G | 17.8 | 79.65 | - | - | - | - | - |
| LTMP-Swin-T (Bonnaerens & Dambre, 2023) | 29.0 M | 3.95 G | 17.9 | - | 79.78 | 79.96 | 80.09 | 80.24 | 80.30 |
| **IBTM-Swin-T (Fine-tuned)** | 29.0 M | 3.82 G | 17.0 | - | **80.06** | **80.46** | **80.79** | **81.20** | **81.38** |
| Swin-B (Liu et al., 2021) | 88.0 M | 15.4 G | 33.9 | 83.50 | - | - | - | - | - |
| ToMe-Swin-B (Bolya et al., 2023) | 88.0 M | 13.0 G | 29.9 | 81.87 | - | - | - | - | - |
| ToFu-Swin-B (Kim et al., 2024) | 88.0 M | 13.0 G | 30.1 | 82.04 | - | - | - | - | - |
| LTMP-Swin-B (Bonnaerens & Dambre, 2023) | 88.0 M | 13.2 G | 30.4 | - | 82.24 | 82.39 | 82.45 | 82.51 | 82.55 |
| **IBTM-Swin-B (Fine-tuned)** | 88.0 M | 12.0 G | 29.6 | - | **82.50** | **82.72** | **82.88** | **83.43** | **83.64** |
| ViT-S (Dosovitskiy et al., 2021a) | 22.1 M | 4.30 G | 22.5 | 81.20 | - | - | - | - | - |
| ToMe-ViT-S (Bolya et al., 2023) | 22.1 M | 3.82 G | 18.4 | 80.04 | - | - | - | - | - |
| ToFu-ViT-S (Kim et al., 2024) | 22.1 M | 3.82 G | 18.7 | 80.19 | - | - | - | - | - |
| LTMP-ViT-S (Bonnaerens & Dambre, 2023) | 22.1 M | 3.89 G | 19.0 | - | 80.32 | 80.40 | 80.35 | 80.41 | 80.50 |
| **IBTM-ViT-S (Fine-tuned)** | 22.1 M | 3.70 G | 18.2 | - | **80.47** | **80.69** | **80.94** | **81.27** | **81.55** |
| ViT-B (Dosovitskiy et al., 2021a) | 86.5 M | 17.58 G | 37.2 | 83.74 | - | - | - | - | - |
| ToMe-ViT-B (Bolya et al., 2023) | 86.5 M | 13.12 G | 31.0 | 82.86 | - | - | - | - | - |
| ToFu-ViT-B (Kim et al., 2024) | 86.5 M | 13.12 G | 31.5 | 83.22 | - | - | - | - | - |
| LTMP-ViT-B (Bonnaerens & Dambre, 2023) | 86.5 M | 13.46 G | 32.7 | - | 83.29 | 83.40 | 83.44 | 83.50 | 83.55 |
| **IBTM-ViT-B (Fine-tuned)** | 86.5 M | 12.85 G | 30.7 | - | **83.35** | **83.57** | **83.76** | **83.91** | **83.96** |

Table 1: Performance comparison between IBTM with competing token merging baselines, ToMe (Bolya et al., 2023), ToFu (Kim et al., 2024), and LTMP (Bonnaerens & Dambre, 2023) in fine-tuning setup on ImageNet. Among the compared methods, ToMe and ToFu do not require training. Both IBTM models and LTMP models are fine-tuned for 1, 5, 10, 25, and 50 epochs for fair comparisons.

In addition, we also compare the IBTM with three token merging methods, ToMe (Bolya et al., 2023), ToFu (Kim et al., 2024), and LTMP (Bonnaerens & Dambre, 2023), to demonstrate the superiority of our IBTM. The results are shown in Table 1. The inference time of all the models is also evaluated on the validation set of ImageNet-1k and reported in milliseconds (ms) per batch for an evaluation batch size of 128 on one Nvidia A100 GPU. We set the compression ratio of all the competing token merging methods to 0.75 to demonstrate that our IBTM-Transformers render higher top-1 accuracy with even fewer FLOPs and fatter inference speed compared to the current state-of-the-art token merging methods. It is observed that our IBTM models under the fine-tuning setup achieve significantly better prediction accuracy with fewer FLOPs and inference time compared to the LTMP models fine-tuned for the same number of training epochs. For example, IBTM-MobileViT-S, fine-tuned for 50 epochs, outperforms the LTMP-MobileViT-S, which is also fine-tuned for 50 epochs by $0.89\%$ in top-1 accuracy with less FLOPs and faster inference speed. In Section B.4, we further compare the IBTM with the state-of-the-art token pruning methods in the fine-tuning setup.

### 4.1.2 TRAINING-FROM-SCRATCH SETUP

In the training-from-scratch setup, all the parameters, including the pre-trained weights and the weights $\{M^{(\ell)}\}$ in the token merging mask modules, of IBTM models are updated in the training process. To train IBTM-Transformers from scratch, we utilize the AdamW optimizer with $\beta_1 = 0.9$ and $\beta_2 = 0.999$. The training process spans 300 epochs, starting with a warm-up phase during

which token merging is not applied in all the IBTM blocks. After the warm-up stage, we enable token merging in all the IBTM blocks. $t_{\text{warm}}$ is fixed to 100 in all the experiments. We set the weight decay at 0.01. The learning rate initially increases from 0.0002 to 0.002 over the first 10 epochs and is subsequently reduced back to 0.0002 following a cosine decay schedule.

The results are deferred to Table 3 in Section B.1 of the appendix. It is observed from the results that models integrated with IBTM show reduced FLOPs and enhanced accuracy compared to their original vision transformer counterparts. For instance, IBTM-MobileViT-S not only reduces its FLOPs from 1.4G to 1.17G but also improves accuracy by $1.3\%$ over the original MobileViT-S. To further demonstrate the efficiency of the IBTM, we compare it against current state-of-the-art weight pruning methods for efficient vision transformers, including $S^2$ViTE (Chen et al., 2021b), SPViT (Kong et al., 2022b), and SAViT (Zheng et al., 2022b) on EfficientViT-B1 (r224). To apply $S^2$ViTE, SPViT, and SAViT on EfficientViT-B1 (r224), we first run their pruning process following the standard implementation in their papers (Chen et al., 2021b; Kong et al., 2022b; Zheng et al., 2022b) on the ImageNet training data. After obtaining the pruned networks, we fine-tune them using the same setting as in (Cai et al., 2023). It is observed from the results that with even lower FLOPs, IBTM models trained from scratch consistently outperform the competing baseline methods.

## 4.2 ABLATION STUDY

**Study on the effectiveness of IBTM in reducing IB loss.** We study the effectiveness of IBTM in reducing the IB loss and the variational upper bound of IB loss, which is the IB bound, across three vision transformers, including MobileViT-S, MobileViT-XS, and EfficientViT (r224). We compare the performance of the vision transformers with the baseline token merging method, ToME (Bolya et al., 2023), LTMP (Bonnaerens & Dambre, 2023), and the corresponding IBTM-Tranformer models with all the transformer blocks replaced with the IBTM blocks. The ablation study results for the fine-tuning setup are shown in Table 2. The ablation study results for the train-from-scratch setup are deferred to Table 7 in Section E.1 of the appendix, respectively. The results indicate that although ToMe and LTMP reduce the IB loss and the IB bound in both the fine-tuning setup and the train-from-scratch setup, thereby adhering to the IB principle, which aims to enhance the correlation of features with class labels while reducing their correlation with the input, IBTM can further decrease the IB loss and IB bound. In particular, our IBTM models improve the vanilla vision transformers, the ToMe models, and the LTMP models by a large margin in terms of both IB loss and top-1 accuracy for both the fine-tuning setup and the train-from-scratch setup.

| Model | FLOPs | Top-1 | | | | IB Bound | | | | IB Loss | | | |
|---|---|---|---|---|---|---|---|---|---|---|---|---|---|
| | | 0 | 1 | 10 | 50 | 0 | 1 | 10 | 50 | 0 | 1 | 10 | 50 |
| MobileViT-S | 1.40 G | 78.40 | - | - | - | 0.05782 | - | - | - | -0.00432 | - | - | - |
| ToMe-MobileViT-S | 1.22 G | 76.72 | - | - | - | 0.04931 | - | - | - | -0.00525 | - | - | - |
| LTMP-MobileViT-S | 1.26 G | - | 77.53 | 77.82 | 78.14 | - | 0.04902 | 0.04735 | 0.04542 | - | -0.00765 | -0.00874 | -0.00913 |
| **IBTM-MobileViT-S** | 1.17 G | - | **77.72** | **78.34** | **79.05** | - | **0.03095** | **0.02967** | **0.02683** | - | **-0.01430** | **-0.01576** | **-0.01692** |
| EfficientViT-B1 | 0.52 G | 79.40 | - | - | - | 0.06014 | - | - | - | -0.00451 | - | - | - |
| ToMe-EfficientViT-B1 | 0.47 G | 78.81 | - | - | - | 0.04642 | - | - | - | -0.00732 | - | - | - |
| LTMP-EfficientViT-B1 | 0.52 G | - | 79.21 | 79.32 | 79.40 | - | 0.04537 | 0.04219 | 0.03970 | - | -0.00802 | -0.00916 | -0.00995 |
| **IBTM-EfficientViT-B1** | 0.44 G | - | **79.39** | **79.62** | **80.22** | - | **0.02874** | **0.02703** | **0.02635** | - | **-0.01585** | **-0.01664** | **-0.01704** |
| ViT-B | 17.58 G | 83.74 | - | - | - | 0.05539 | - | - | - | -0.00419 | - | - | - |
| ToMe-ViT-B | 13.12 G | 82.86 | - | - | - | 0.04583 | - | - | - | -0.00647 | - | - | - |
| LTMP-ViT-B | 13.46 G | - | 83.29 | 83.44 | 83.55 | - | 0.04392 | 0.04275 | 0.04086 | - | -0.00665 | -0.00693 | -0.00752 |
| **IBTM-ViT-B** | 12.85 G | - | **83.35** | **83.76** | **83.96** | - | **0.03732** | **0.03506** | **0.03082** | - | **-0.01425** | **-0.01572** | **-0.01618** |

Table 2: Ablation study on the effects of IBTM in reducing IB loss in the fine-tuning setup. Both LTMP models and IBTM models fine-tuned for 1, 10, 50 epochs are evaluated.

**Study on the IB loss and IB bound at different layers.** To study how the IB loss $\text{IB}(G)$, and the variational upper bound for the IB loss, $\text{IBU}(G)$, decrease with respect to layer index $\ell$ of an IBTM network, $\text{IB}(G)$ and $\text{IBU}(G)$ across different transformer layers for both LTMP-MobileViT-S and IBTM-MobileViT-S trained in the fine-tuning setting are illustrated in Figure 2. Both models contain 9 transformer layers. It is observed from Figure 2 that both $\text{IB}(G)$ and $\text{IBU}(G)$ decrease in deeper layers with larger layer indices of LTMP-MobileViT-S and IBTM-MobileViT-S. This observation suggests that features in deeper layers correlate more closely with the class labels and less with the input features, adhering to the IB principle. Moreover, IBTM-MobileViT-S reduces both $\text{IB}(G)$ and $\text{IBU}(G)$ to lower levels in deeper layers compared to LTMP-MobileViT-S. These observations evidence that the mask module in the IBTM block which generates the informative token merging task by (3) can effectively reduce both $\text{IB}(G)$ and $\text{IBU}(G)$, better adhering to the IB principle than the baseline LTMP-MobileViT-S.

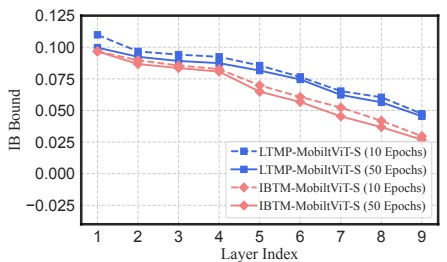 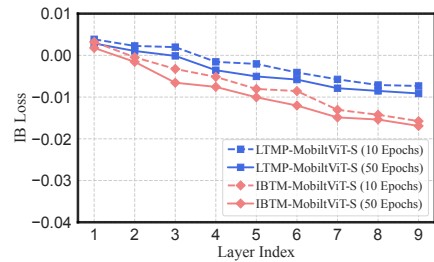

(a) IB bound ($\text{IBU}(G)$) comparison between LTMP-MobileViT-S and IBTM-MobileViT-S.

(b) IB loss ($\text{IB}(G)$) comparison between LTMP-MobileViT-S and IBTM-MobileViT-S.

Figure 2: IB bound and IB loss comparison between MobileViT-S and IBTM-MobileViT-S at different transformer layers.

Figure 3 in the appendix illustrates the training loss and the test loss during the training process of IBTM-MobileViT-S, highlighting that the test loss of the IBTM network exhibits a more rapid decline compared to the vanilla MobileViT-S. We also compare the training time of IBTM models with the competing baselines for token merging in Table 9 in the appendix.

## 5 CONCLUSION

In this paper, we propose a novel transformer block, Transformer with Information Bottleneck inspired Token Merging, or IBTM. IBTM blocks perform token merging so as to render a transformer network with less FLOPs and faster inference speed. An IBTM block generates an informative token merging mask for token merging in a learnable manner, which is inspired by the reduction of the Information Bottleneck (IB) loss. A network with IBTM blocks can be trained from scratch or fine-tuned from a pre-trained backbone with standard SGD, and it enjoys a reduction of IB loss and reduced FLOPs while maintaining a compelling prediction accuracy. We demonstrate the effectiveness of IBTM by replacing all the transformer blocks in several popular vision transformers with IBTM blocks. Extensive experiments on various computer vision tasks demonstrate the effectiveness of IBTM.

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

## A  THE INFORMATION BOTTLENECK (IB) PERSPECTIVE OF THE TOKEN MERGING

Such an idea of informative token merging can also be viewed from the perspective of Information Bottleneck (IB). Let $Z$ be the original attention output tokens, which are merged into the merged tokens denoted by $\tilde{X}$, and let $Y$ be the ground truth training labels for a classification task. $\tilde{X}$ has less tokens than $Z$. The principle of IB is to maximize the mutual information between $\tilde{X}$ and $Y$ while minimizing the mutual information between $\tilde{X}$ and $X$. That is, IB encourages the network to learn the merged tokens more correlated with the class labels while reducing their correlation with the input. Extensive empirical and theoretical works have evidenced that models respecting the IB principle enjoy compelling generalization. With the informative token merging process in IBTM, the merged tokens $\tilde{X}$ are the informative aggregation of the original attention output tokens $Z$, so $\tilde{X}$ are less correlated with the training images and in this manner the IB principle is better adhered. This is reflected in Table 2 in Section 4.2 and Table 7 in Section E.1 of the appendix, where models for ablation study with existing token merging methods, ToMe (Bolya et al., 2023) and LTMP (Bonnaerens & Dambre, 2023), enjoys less IB loss than the corresponding vanilla transformers. This observation indicates that the IB principle is better respected by the token merging process in ToMe and LTMP. In order to further decrease the IB loss, we propose an Information Bottleneck (IB) inspired token merging process, where a IBTM block generates an informative token merging task which reduces the IB loss for vision transformers. For example, our model termed "IBTM-MobileViT-S" in Table 2 in Section 4.2 and Table 7 in Section E.1 of the appendix is the vision transformer with the IB loss reduced by replacing all the transformer blocks in MobileViT-S with IBTM blocks so that more informative merged tokens are generated by the proposed informative token merging process. While ToMe and LTMP hurts the prediction accuracy compared to the vanilla model, our IBTM enjoys even higher top-1 accuracy than the vanilla MobileViT-S either trained from scratch or fine-tuned from pre-trained checkpoints, and we have the same observations for MobileViT-XS and EfficientViT.

## B  MORE EXPERIMENTAL RESULTS

### B.1  IMAGENET CLASSIFICATION RESULTS FOR THE TRAINING-FROM-SCRATCH SETUP

The ImageNet classification results of IBTM models trained from scratch are shown in Table 3.

### B.2  OBJECT DETECTION

**Implementation details.** We incorporate ImageNet pre-trained models, that are IBTM-MobileViT-XS, IBTM-MobileViT-S, and IBTM-EfficientViT, with the single-shot object detection backbone, SSDLite (Sandler et al., 2018), to evaluate on the MS-COCO dataset (Lin et al., 2014), which comprises 117k training images and 5k validation images. We fine-tune all pre-trained IBTMs within the object detection framework at a standard input resolution of $320 \times 320$. These models undergo a training period of 200 epochs using the AdamW optimizer, adhering to the training protocols established in (Mehta & Rastegari, 2022). Employing a cosine learning rate scheduler, the initial learning rate of $0.0009$ is gradually reduced to $1.6e^{-6}$. For the object localization, we utilize a smooth $\ell^1$ loss, and for classification, cross-entropy losses are applied. The evaluation of performance on the validation set is conducted using the mAP metric with an IoU range from 0.50 to 0.95 in increments of 0.05.

**Results.** We adopt a comparative study of our IBTM Transformers against other lightweight feature backbones within the SSDLite object detection framework. The results, as detailed in Table 4 of the appendix, illustrate significant improvements in object detection performance when the feature backbone is upgraded to include IBTM blocks. For example, substituting MobileViT-S with IBTM-MobileViT-S enhances the mAP by $0.7\%$ while concurrently reducing FLOPs by $0.3$G. Additionally, SSDLite equipped with IBTM-EfficientViT achieves a substantial performance increase of $6.9\%$ while maintaining the same FLOPs as MobileNetV3.

| Model | # Params | FLOPs | Top-1 |
|---|---|---|---|
| MobileViT-XS | 2.3 M | 0.7 G | 74.8 |
| ToMe-MobileViT-XS (Bolya et al., 2023) | 2.3 M | 0.54 G | 72.7 |
| ToFu-MobileViT-XS (Kim et al., 2024) | 2.3 M | 0.54 G | 73.3 |
| LTMP-MobileViT-XS (Bonnaerens & Dambre, 2023) | 2.3 M | 0.56 G | 73.9 |
| **IBTM-MobileViT-XS (Ours)** | 2.3 M | 0.52 G | **75.8** |
| Mobile-Former | 9.4 M | 0.2 G | 76.7 |
| EfficientFormer (Li et al., 2022) | 12.3 M | 1.3 G | 79.2 |
| MobileViT-S | 5.6 M | 1.4 G | 78.4 |
| ToMe-MobileViT-S (Bolya et al., 2023) | 5.6 M | 1.22 G | 76.7 |
| ToFu-MobileViT-S (Kim et al., 2024) | 5.6 M | 1.22 G | 77.2 |
| LTMP-MobileViT-S (Bonnaerens & Dambre, 2023) | 5.6 M | 1.26 G | 77.5 |
| **IBTM-MobileViT-S (Ours)** | 5.6 M | 1.17 G | **79.7** |
| EfficientViT-B1 [r224] (Cai et al., 2023) | 9.1 M | 0.52 G | 79.4 |
| $S^2$ViTE-EfficientViT-B1 [r224] (Chen et al., 2021b) | 8.2 M | 0.47 G | 79.0 |
| SPViT-EfficientViT-B1 [r224] (Kong et al., 2022b) | 9.2 M | 0.49 G | 79.3 |
| SAViT-EfficientViT-B1 [r224] (Zheng et al., 2022b) | 8.4 M | 0.47 G | 79.2 |
| ToMe-EfficientViT-B1 [r224] (Bolya et al., 2023) | 9.1 M | 0.47 G | 78.8 |
| ToFu-EfficientViT-B1 [r224] (Kim et al., 2024) | 9.1 M | 0.47 G | 79.0 |
| LTMP-EfficientViT-B1 [r224] (Bonnaerens & Dambre, 2023) | 9.1 M | 0.50 G | 79.2 |
| **IBTM-EfficientViT-B1 [r224] (Ours)** | 9.1 M | 0.44 G | **80.2** |
| EfficientViT-B1 [r288] (Cai et al., 2023) | 9.1 M | 0.86 G | 80.4 |
| ToMe-EfficientViT-B1 [r288] (Bolya et al., 2023) | 9.1 M | 0.73 G | 79.7 |
| ToFu-EfficientViT-B1 [r288] (Kim et al., 2024) | 9.1 M | 0.73 G | 79.8 |
| LTMP-EfficientViT-B1 [r288] (Bonnaerens & Dambre, 2023) | 9.1 M | 0.76 G | 80.0 |
| **IBTM-EfficientViT-B1 [r288] (Ours)** | 9.1 M | 0.70 G | **81.0** |
| ViT-S/16 (Dosovitskiy et al., 2021b) | 22.1 M | 4.3 G | 81.2 |
| **IBTM-ViT-S/16 (Ours)** | 22.1 M | 3.7 G | **81.8** |
| ViT-B/16 (Dosovitskiy et al., 2021b) | 22.1 M | 4.3 G | 81.2 |
| **IBTM-ViT-S/16 (Ours)** | 22.1 M | 3.7 G | **81.8** |
| Swin-T (Liu et al., 2021) | 29.0 M | 4.5 G | 81.3 |
| **IBTM-Swin-T (Ours)** | 29.0 M | 3.8 G | **81.8** |
| Swin-B (Liu et al., 2021) | 29.0 M | 4.5 G | 81.3 |
| **IBTM-Swin-B (Ours)** | 29.0 M | 3.8 G | **81.8** |

Table 3: Comparisons with baseline methods on ImageNet-1k validation set.

| Feature backbone | # Params. | FLOPs | mAP |
|---|---|---|---|
| MobileNetv3 (Howard et al., 2019) | 4.9 M | 1.4 G | 22.0 |
| MobileNetv2 (Sandler et al., 2018) | 4.3 M | 1.6 G | 22.1 |
| MobileNetv1 (Howard et al., 2017) | 5.1 M | 2.6 G | 22.2 |
| MixNet (Tan & Le, 2019) | 4.5 M | 2.2 G | 22.3 |
| MNASNet (Tan et al., 2019) | 4.9 M | 1.7 G | 23.0 |
| YoloV5-N (640×640) (Redmon & Farhadi, 2017) | 1.9 M | 4.5 G | 28.0 |
| Vidt (Song et al., 2022) | 7.0 M | 6.7 G | 28.7 |
| MobileViT-XS | 2.7 M | 1.7 G | 24.8 |
| **IBTM-MobileViT-XS(Ours)** | 2.7 M | 1.5 G | **25.4** |
| MobileViT-S | 5.7 M | 2.4 G | 27.7 |
| **IBTM-MobileViT-S(Ours)** | 5.7 M | 2.1 G | **28.4** |
| EfficientViT | 9.9 M | 1.5 G | 28.4 |
| **IBTM-EfficientViT(Ours)** | 9.9 M | 1.4 G | **28.9** |

Table 4: Object detection performance with SSDLite.

## B.3 INSTANCE SEGMENTATION

In this section, we assess the efficacy of IBTM when applied to instance segmentation tasks using the COCO dataset (Lin et al., 2014). We utilize Mask R-CNN (He et al., 2017) equipped with a Feature Pyramid Network (FPN) as the segmentation head, built on the IBTM-EfficientViT-B1 feature backbone. For comparative analysis, we include EfficientViT-B1 (Cai et al., 2023) and EViT (Liu et al., 2023) as baseline models. Both our models and the baselines are trained on the training split of the COCO dataset and evaluated on the validation split, adhering to the protocols established by (Chen et al., 2019). The training duration is set to 12 epochs, consistent with the $1\times$ schedule described in (Chen et al., 2019). The AdamW optimizer is employed for training following the practices of (Liu et al., 2023). We initiate the learning rate at $0.001$, which is then gradually reduced following a cosine learning rate schedule. Performance metrics reported include the mean bounding box Average Precision ($mAP^b$) and mean mask Average Precision ($mAP^m$), along with bounding box Average Precision ($AP^b$) and mask Average Precision ($AP^m$) at IoU thresholds of 0.5 and 0.75. The findings, detailed in Table 5, demonstrate that IBTM-EfficientViT-B1 consistently enhances segmentation performance across various thresholds.

| Methods | mAP$^{box}$ | AP$_{50}^b$ | AP$_{75}^b$ | mAP$^m$ | AP$_{50}^m$ | AP$_{75}^m$ |
|---|---|---|---|---|---|---|
| EViT (Liu et al., 2023) | 32.8 | 54.4 | 34.5 | 31.0 | 51.2 | 32.2 |
| EfficientViT-B1 (Cai et al., 2023) | 33.5 | 55.4 | 34.8 | 31.9 | 52.3 | 32.7 |
| IBTM-EfficientViT-B1 | **34.3** | **56.1** | **35.2** | **32.8** | **52.8** | **33.1** |

Table 5: Instance Segmentation Results on COCO.

### B.4 COMPARISONS WITH TOKEN PRUNING METHODS

We compare LTMP with the token pruning methods EViT Liang et al. (2022) and ATS Fayyaz et al. (2022). For a fair comparison, we apply EViT and ATS on the same pre-trained ViT-B backbone used in Table 1 in Section 4.1.1, and apply the same data augmentation strategies as reported in Section 4.1.1. As both EViT and ATS require fine-tuning, we follow the settings in Section 4.1.1 and fine-tune the models compressed with EViT and ATS, that are EViT-ViT-B and ATS-ViT-B for 1, 5, 10, 25, and 50 epochs, respectively, and compare them with IBTM models fine-tuned for the same number of epochs. The results are shown in Table 6. It is observed that with an even faster inference speed, the IBTM-ViT-B achieves higher top-1 accuracy on ImageNet compared to the state-of-the-art token pruning methods. For example, IBTM-ViT-B outperforms EViT-ViT-B by $0.37\%$ in top-1 accuracy after fine-tuning for 50 epochs, demonstrating the superiority of the IBTM in compressing vision transformers.

| Methods | # Params. | Inference Time (ms/batch) | Top-1 Accuracy (%) | | | | | |
|---|---|---|---|---|---|---|---|---|
| | | | 0 | 1 | 5 | 10 | 25 | 50 |
| ViT-B (Dosovitskiy et al., 2021a) | 86.5 M | 37.2 | 83.74 | - | - | - | - | - |
| EViT-ViT-B (Liang et al., 2022) | 86.5 M | 31.9 | - | 82.22 | 82.54 | 83.15 | 83.49 | 83.62 |
| ATS-ViT-B (Fayyaz et al., 2022) | 86.5 M | 31.2 | - | 82.49 | 82.85 | 83.05 | 83.32 | 83.38 |
| **IBTM-ViT-B (Fine-tuned)** | 86.5 M | 30.7 | - | **83.35** | **83.57** | **83.76** | **83.91** | **83.96** |

Table 6: Performance comparison between IBTM and token pruning methods in fine-tuning setup on ImageNet. Both IBTM models and LTMP models are fine-tuned for 1, 5, 10, 25, and 50 epochs for fair comparisons.

## C  PROOFS

### C.1  PROOF OF PROPOSITION 3.2

*Proof.* We first compute the gradient of $\tau(\tilde{X}_i(G), a')$ with respect to $\tilde{X}_i(G)$ by

$$\nabla_{\tilde{X}_i(G)}\tau(\tilde{X}_i(G), a') = \frac{2S_{ia'}\sum\limits_{a=1}^{C} S_{ia}\left(\tilde{X}_i(G) - \tilde{\mathcal{C}}_a\right) - 2S_{ia'}\left(\tilde{X}_i(G) - \tilde{\mathcal{C}}_{a'}\right)\sum\limits_{a=1}^{C} S_{ia}}{\left(\sum\limits_{a=1}^{C} S_{ia}\right)^2}.$$

Using the definitions of $\gamma_i$ and $zeta_i$ as $\gamma_i \coloneqq \sum\limits_{a=1}^{C} S_{ia}$ and $\zeta_i \coloneqq \sum\limits_{a=1}^{C} S_{ia}\mathcal{C}_a$ for $i \in [n]$, we have

$$\nabla_{\tilde{X}_i(G)}\tau(\tilde{X}_i(G), a') = \frac{2S_{ia'}}{\gamma_i^2}\left(\gamma_i\tilde{\mathcal{C}}_{a'} - \zeta_i\right).$$

As a result, the gradient of $\mathrm{IBU}(G)$ with respect to $G$ is computed as follows:

$$
\begin{aligned}
\nabla_G \mathrm{IBU}(G) &= \frac{1}{n} \sum_{i=1}^{n} \sum_{a=1}^{C} \sum_{b=1}^{C} Z_i \nabla_{\tilde{X}_i} \tau(\tilde{X}_i, a) \tau(X_i, b) \log \tau(X_i, b) \\
&\quad - \frac{1}{n} \sum_{i=1}^{n} \sum_{a=1}^{C} \sum_{y=1}^{C} Z_i \nabla_{\tilde{X}_i} \tau(\tilde{X}_i, a) \mathbb{1}_{\{y_i = y\}} \log Q(\tilde{X} \in a | Y = y) \\
&= \frac{1}{n} \sum_{i=1}^{n} \sum_{a=1}^{C} Z_i \nabla_{\tilde{X}_i} \tau(\tilde{X}_i, a) \psi_{i,a} \\
&= \frac{2}{n} \sum_{i=1}^{n} \sum_{a=1}^{C} Z_i \frac{S_{ia}}{\gamma_i^2} \left( \gamma_i \mathcal{C}_a - \zeta_i \right) \psi_{i,a},
\end{aligned}
$$

where $\psi_{i,a} := \sum_{b=1}^{C} \tau(X_i, b) \log \tau(X_i, b) - \sum_{y=1}^{C} \mathbb{1}_{\{y_i = y\}} \log Q(\tilde{X} \in a | Y = y)$.

$\square$

## C.2 PROOF OF THEOREM 3.1

We need the following two lemmas before the proof of Theorem 3.1. It is noted that we abbreviate $\tilde{X}(G)$ and $\tilde{X}_i(G)$ as $\tilde{X}$ and $\tilde{X}_i$ in the sequel.

**Lemma C.1.**

$$
I(\tilde{X}, X) \le \frac{1}{n} \sum_{i=1}^{n} \sum_{a=1}^{C} \sum_{b=1}^{C} \tau(\tilde{X}_i, a) \tau(X_i, b) \log \tau(X_i, b) - \frac{1}{n^2} \sum_{i=1}^{n} \sum_{j=1}^{n} \sum_{b=1}^{C} \tau(X_i, b) \log \tau(X_j, b)
$$

(5)

**Lemma C.2.**

$$
I(\tilde{X}, Y) \ge \frac{1}{n} \sum_{a=1}^{C} \sum_{y=1}^{C} \sum_{i=1}^{n} \tau(\tilde{X}_i, a) \mathbb{1}_{\{y_i = y\}} \log Q(\tilde{X} \in a | Y = y)
$$

(6)

***Proof of Theorem 3.1.*** We note that $\mathrm{IB}(\mathcal{W}) = I(\tilde{X}, X) - I(\tilde{X}, Y)$. Then $\mathrm{IB}(\mathcal{W}) \le \mathrm{IBU}(\mathcal{W}) - C_0$ follows by the upper bound for $I(\tilde{X}, X)$ in Lemma C.1 and the lower bound for $I(\tilde{X}, Y)$ in Lemma C.2. Here $C_0 = \frac{1}{n^2} \sum_{i=1}^{n} \sum_{j=1}^{n} \sum_{b=1}^{C} \tau(X_i, b) \log \tau(X_j, b)$.

$\square$

***Proof of Lemma C.1.*** By the log sum inequality, we have

$$I(\tilde{X}, X)$$

$$= \sum_{a=1}^{C} \sum_{b=1}^{C} \Pr\left[\tilde{X} \in a, X \in b\right] \log \frac{\Pr\left[\tilde{X} \in a, X \in b\right]}{\Pr\left[\tilde{X} \in a\right] \Pr\left[X \in b\right]}$$

$$\leq \frac{1}{n^2} \sum_{i=1}^{n} \sum_{j=1}^{n} \sum_{a=1}^{C} \sum_{b=1}^{C} \tau(\tilde{X}_i, a)\tau(X_i, b) \left(\log\left(\tau(\tilde{X}_i, a)\tau(X_i, b)\right)\right.$$

$$\left. - \log\left(\tau(\tilde{X}_i, a)\tau(X_j, b)\right)\right)$$

$$= \frac{1}{n^2} \sum_{i=1}^{n} \sum_{j=1}^{n} \sum_{a=1}^{C} \sum_{b=1}^{C} \tau(\tilde{X}_i, a)\tau(X_i, b) \log \tau(X_i, b)$$

$$- \frac{1}{n^2} \sum_{i=1}^{n} \sum_{j=1}^{n} \sum_{a=1}^{C} \sum_{b=1}^{C} \tau(\tilde{X}_i, a)\tau(X_i, b) \log \tau(X_j, b)$$

$$= \frac{1}{n} \sum_{i=1}^{n} \sum_{a=1}^{C} \sum_{b=1}^{C} \tau(\tilde{X}_i, a)\tau(X_i, b) \log \tau(X_i, b)$$

$$- \frac{1}{n^2} \sum_{i=1}^{n} \sum_{j=1}^{n} \sum_{a=1}^{C} \sum_{b=1}^{C} \tau(\tilde{X}_i, a)\tau(X_i, b) \log \tau(X_j, b). \tag{7}$$

$\square$

**Proof of Lemma C.2.** Let $Q(\tilde{X}|Y)$ be a variational distribution. We have

$$I(\tilde{X}, Y)$$

$$= \sum_{a=1}^{C} \sum_{y=1}^{C} \Pr\left[\tilde{X} \in a, Y = y\right] \log \frac{\Pr\left[\tilde{X} \in a, Y = y\right]}{\Pr\left[\tilde{X} \in a\right] \Pr\left[Y = y\right]}$$

$$= \sum_{a=1}^{C} \sum_{y=1}^{C} \Pr\left[\tilde{X} \in a, Y = y\right] \log \frac{\Pr\left[\tilde{X} \in a | Y = y\right] Q(\tilde{X} \in a | Y = y)}{\Pr\left[\tilde{X} \in a\right] Q(\tilde{X} \in a | Y = y)}$$

$$\geq \sum_{a=1}^{C} \sum_{y=1}^{C} \Pr\left[\tilde{X} \in a, Y = y\right] \log \frac{\Pr\left[\tilde{X} \in a | Y = y\right]}{Q(\tilde{X} \in a | Y = y)}$$

$$+ \sum_{a=1}^{C} \sum_{y=1}^{C} \Pr\left[\tilde{X} \in a, Y = y\right] \log \frac{Q(\tilde{X} \in a | Y = y)}{\Pr\left[\tilde{X} \in a\right]}$$

$$= \mathrm{KL}\left(P(\tilde{X}|Y) \middle\| Q(\tilde{X}|Y)\right)$$

$$+ \sum_{a=1}^{C} \sum_{y=1}^{C} \Pr\left[\tilde{X} \in a, Y = y\right] \log \frac{Q(\tilde{X} \in a | Y = y)}{\Pr\left[\tilde{X} \in a\right]}$$

$$\geq \sum_{a=1}^{C} \sum_{y=1}^{C} \Pr\left[\tilde{X} \in a, Y = y\right] \log \frac{Q(\tilde{X} \in a | Y = y)}{\Pr\left[\tilde{X} \in a\right]}$$

$$= \sum_{a=1}^{C} \sum_{y=1}^{C} \Pr\left[\tilde{X} \in a, Y = y\right] \log Q(\tilde{X} \in a | Y = y) + H\left(P(\tilde{X})\right)$$

$$\geq \sum_{a=1}^{C} \sum_{y=1}^{C} \Pr\left[\tilde{X} \in a, Y = y\right] \log Q(\tilde{X} \in a | Y = y)$$

$$\geq \frac{1}{n} \sum_{a=1}^{C} \sum_{y=1}^{C} \sum_{i=1}^{n} \tau(\tilde{X}_i, a) \mathbb{1}_{\{y_i = y\}} \log Q(\tilde{X} \in a | Y = y). \tag{8}$$

$\square$

### C.3 COMPUTATION OF $Q^{(t)}(\tilde{\mathbf{X}}|Y)$

The variational distribution $Q^{(t)}(\tilde{\mathbf{X}}|Y)$ can be computed by

$$Q^{(t)}(\tilde{X} \in a | Y = y) = \Pr\left[\tilde{X} \in a | Y = y\right]$$

$$= \frac{\sum_{i=1}^{n} \tau(\tilde{X}_i, a) \mathbb{1}_{\{y_i = y\}}}{\sum_{i=1}^{n} \mathbb{1}_{\{y_i = y\}}}. \tag{9}$$

## D   IMPLEMENTATION DETAILS

### D.1   COMPUTATION COST ANALYSIS OF IBTM-EFFICIENTVIT

In this section, we analyze the additional inference computation cost, the FLOPs, of the IBTM transformer block for token merging in both regular transformers and efficient transformers as illustrated in Figure 1. Let $D$ be the dimension of input tokens and $N$ be the number of tokens.

The FLOPs of the token merging in an IBTM transformer block in regular vision transformers is $6CDP + 3C + ND^2 + NDP$, where $6CDP + 3C + ND^2$ is the FLOPs for calculating the merging mask and $NDP$ is the cost for applying the merging mask on the input tokens. In the IBTM transformer block of efficient vision transformers, the additional FLOPs of the token merging is $6CDP + 3C + ND^2 + 2NDP$, since the merging mask will be applied to both the input tokens and the merged tokens.

## D.2 Token Merging on Hierarchical ViTs

For the experiments with hierarchical ViTs such as Swin Transformers (Liu et al., 2021), the token merging is performed between the multi-head self-attention modules with either regular or shifted windows and the MLP layers. As a result, the input to the MLP layers has a reduced number of tokens, thereby decreasing the computational costs in the MLP layers. Following the MLP layers, the merged tokens are padded with the same features to restore the original number of tokens. This ensures compatibility with the hierarchical structure of the Swin Transformer, maintaining alignment with subsequent operations such as patch merging and window-based self-attention. This design allows the token merging process to integrate seamlessly with the Swin Transformer architecture while preserving its structural and computational efficiency.

# E Ablation Study

## E.1 Study on the effects of IBTM in reducing IB loss for the train-from-scratch setup

We also conduct an ablation study on the effects of IBTM in reducing IB loss for the train-from-scratch setup. The results are shown in Table 7.

| Model | FLOPs | Top-1 | IB Bound | IB Loss |
|---|---|---|---|---|
| MobileViT-S | 1.40 G | 78.40 | 0.05782 | -0.00432 |
| ToMe-MobileViT-S | 1.22 G | 76.72 | 0.04931 | -0.00525 |
| LTMP-MobileViT-S | 1.26 G | 78.14 | 0.04542 | -0.00913 |
| IBTM-MobileViT-S | 1.17 G | **79.68** | **0.02425** | **-0.01725** |
| EfficientViT-B1 | 0.52 G | 79.40 | 0.06014 | -0.00451 |
| ToMe-EfficientViT-B1 | 0.47 G | 78.81 | 0.04642 | -0.00732 |
| LTMP-EfficientViT-B1 | 0.52 G | 79.40 | 0.03970 | -0.00995 |
| IBTM-EfficientViT-B1 | 0.44 G | **80.20** | **0.02689** | **-0.01730** |
| ViT-B | 17.58 G | 83.74 | 0.05539 | -0.00419 |
| ToMe-ViT-B | 13.12 G | 82.86 | 0.04583 | -0.00647 |
| LTMP-ViT-B | 13.46 G | 83.55 | 0.04086 | -0.00752 |
| IBTM-ViT-B | 12.85 G | **83.87** | **0.03094** | **-0.01636** |

Table 7: Ablation Study on the effects of IBTM in reducing IB loss in the train-from-scratch setup.

## E.2 Study on the Impact of Compression Ratio

We also conduct an ablation study on the compression ratio of token merging on ViT-B. The inference time of all the models is also evaluated on the validation set of ImageNet-1k and reported in milliseconds (ms) per batch for an evaluation batch size of 128 on one Nvidia A100 GPU. It is observed from the results in Table 8 that although a smaller compression ratio can result in a slight accuracy drop, the IBTM-ViT-B with a compression ratio of 0.65 can still achieve the same performance as the original ViT-B model.

## E.3 Training Time Evaluation

We evaluate the training cost of our IBTM models and the baseline models on the training set of ImageNet-1k. The training is performed on 4 NVIDIA A100 GPUs with an effective batch size of 512 images. We report the overall training time of 300 epochs. We also include the training

| Methods | FLOPs (G) | Inference Time (ms/batch) | Compression Ratio $r$ | Train-from-scratch | Fine-tuning |
|---|---|---|---|---|---|
| ViT-B | 17.58 | 37.2 | 1.00 | 83.74 | 83.74 |
| IBTM-ViT-B | 16.55 | 36.5 | 0.95 | 84.43 | 84.32 |
| IBTM-ViT-B | 15.25 | 35.4 | 0.90 | 84.46 | 84.29 |
| IBTM-ViT-B | 14.19 | 34.1 | 0.85 | 84.33 | 84.35 |
| IBTM-ViT-B | 14.89 | 33.5 | 0.80 | 84.15 | 84.20 |
| IBTM-ViT-B | 13.49 | 32.8 | 0.75 | 83.95 | 84.05 |
| IBTM-ViT-B | 12.85 | 31.4 | 0.70 | 83.87 | 83.96 |
| IBTM-ViT-B | 11.95 | 29.6 | 0.65 | 83.74 | 83.81 |
| IBTM-ViT-B | 11.03 | 28.2 | 0.60 | 83.53 | 83.56 |
| IBTM-ViT-B | 10.15 | 27.4 | 0.55 | 83.07 | 83.14 |
| IBTM-ViT-B | 9.63 | 26.3 | 0.50 | 82.87 | 82.95 |
| IBTM-ViT-B | 8.77 | 25.7 | 0.45 | 83.46 | 83.51 |
| IBTM-ViT-B | 8.30 | 24.2 | 0.40 | 82.23 | 82.37 |

Table 8: Performance compression between IBTM-ViT-B with different compression ratios.

time of ToMe Bolya et al. (2023), ToFu (Kim et al., 2024), and LTMP (Bonnaerens & Dambre, 2023) for comparison. It is noted that ToMe, ToFu, and LTMP are applied to pre-trained models. Therefore, the training time for ToMe, ToFu, and LTMP includes the training time of the baseline models. In contrast, our models are trained from scratch. The training time of various models are shown in Table 9. The training overhead of IBTMs mainly comes from the computation of $\left\{\tau(\tilde{X}_i, a)\right\}_{i \in \mathcal{B}_j, a \in [C]}$, $\left\{Q^{(t-1)}(\tilde{X} \in a | Y = y)\right\}_{a \in [C], y \in [C]}$, and $\left\{\mathcal{C}_a^{(t-1)}\right\}_{a=1}^{C}$ as described in Algorithm 1. It is observed from the Table 9 that the training time of IBTM models is comparable to the training time of the competing token merging methods. In addition, IBTM largely resolves the issue of significant prediction accuracy drops after token merging by ToMe, ToFu, and LTMP.

| Methods | # Params | FLOPs | Training Time (Hours) | Top-1 |
|---|---|---|---|---|
| MobileViT-XS | 2.3 M | 0.70 G | 73.5 | 75.8 |
| ToMe-MobileViT-XS | 2.3 M | 0.54 G | 73.5 | 72.7 |
| ToFu-MobileViT-XS | 2.3 M | 0.54 G | 73.5 | 73.3 |
| LTMP-MobileViT-XS | 2.3 M | 0.56 G | 73.8 | 73.9 |
| IBTM-MobileViT-XS | 2.5 M | 0.52 G | 91.0 | **76.8** |
| MobileViT-S | 5.6 M | 1.40 G | 89.5 | 78.4 |
| ToMe-MobileViT-S | 5.6 M | 1.22 G | 89.5 | 76.7 |
| ToFu-MobileViT-S | 5.6 M | 1.22 G | 89.5 | 77.2 |
| LTMP-MobileViT-S | 5.6 M | 1.17 G | 90.0 | 77.5 |
| IBTM-MobileViT-S | 5.9 M | 1.22 G | 105.0 | **79.7** |
| EfficientViT-B1 [r224] | 9.1 M | 0.52 G | 73.0 | 79.4 |
| ToMe-EfficientViT-B1 [r224] | 9.1 M | 0.47 G | 73.0 | 78.8 |
| ToFu-EfficientViT-B1 [r224] | 9.1 M | 0.47 G | 73.0 | 79.0 |
| LTMP-EfficientViT-B1 [r224] | 9.1 M | 0.50 G | 73.3 | 79.2 |
| IBTM-EfficientViT-B1 [r224] | 9.5 M | 0.44 G | 91.0 | **80.2** |
| EfficientViT-B1 [r288] | 9.1 M | 0.86 G | 95.5 | 80.4 |
| ToMe-EfficientViT-B1 [r288] | 9.1 M | 0.73 G | 95.5 | 79.7 |
| ToFu-EfficientViT-B1 [r288] | 9.1 M | 0.73 G | 95.5 | 79.8 |
| LTMP-EfficientViT-B1 [r288] | 9.1 M | 0.76 G | 95.9 | 80.0 |
| IBTM-EfficientViT-B1 [r288] | 9.5 M | 0.70 G | 110.5 | **81.0** |

Table 9: Training time (minutes/epoch) comparisons between IBTMs and their baseline models.

### E.4 TRAINING LOSS AND TEST LOSS OF IBTM-TRANSFOMERS

In this section, we illustrate the training loss and the test loss of IBTM-MobileViT-S. In comparison, we also illustrate the training loss and test loss of MobileViT-S. Both models are trained for 300 epochs. The plots are shown in Figure 3. It can be observed that IBTM-MobileViT-S leads to a lower training loss and test loss at the end of the training, which demonstrates the benefit of IBTM in improving the performance of the vision transformers through the IB-inspired token merging.

### E.5 VISIONIZATION RESULTS

To study the effectiveness of IBTM in selecting informative tokens during the token merging process, we visionize the token merging masks in the first IBTM block of IBTM-MobileViT-S for selected images from ImageNet in Figure 4. Each image is divided into $16 \times 16$ tokens. For each example, we select only the most representative merged token that encapsulates the critical features of the objects

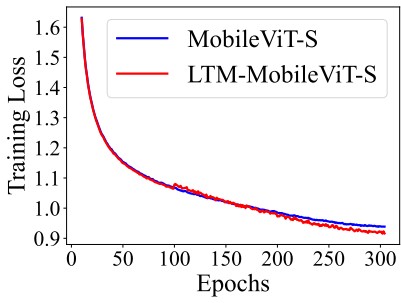 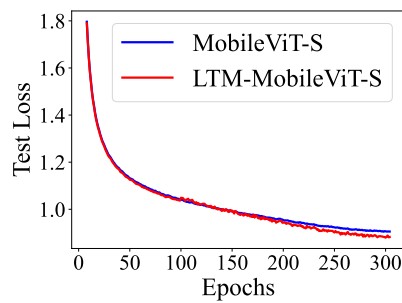

(a) Training loss comparison between MobileViT-S and IBTM-MobileViT-S.

(b) Test loss comparison between MobileViT-S and IBTM-MobileViT-S.

Figure 3: Training loss and test loss comparison between MobileViT-S and IBTM-MobileViT-S.

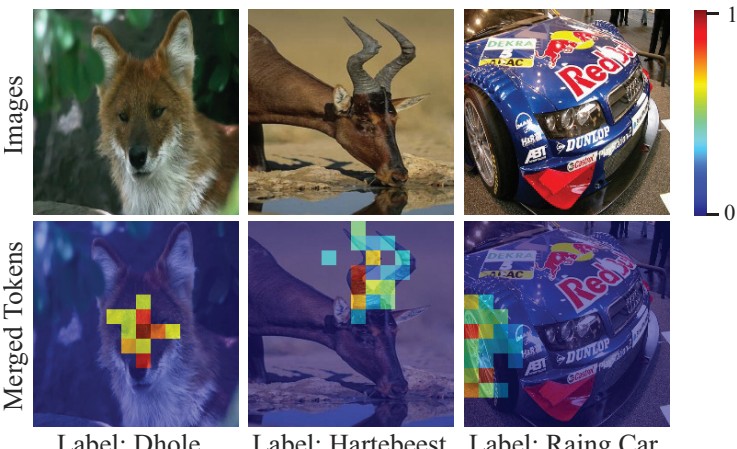

Figure 4: visionization of merging weights in the first IBTM block in IBTM-MobileViT-S.

in the image, and the merged token is a weighted average of several self-attention output tokens with the aggregation weights in the token merging mask. The input images are illustrated in the first row, and the heatmaps that visionize the aggregation weights in the token merging mask for the selected merged token are shown in the second row. The class labels for each image are presented at the bottom of each column. The results illustrate that the mask module in the IBTM block usually assigns higher aggregation weights to tokens covering the most representative and distinctive parts of the objects, which are often the most informative for classifying the images. In the example of the dhole in the first column, the IBTM block puts larger weights on the eyes and nose of the Dhole. In the example of the hartebeest in the second column, the IBTM block puts larger weights on the twisted horns of the hartebeest. In the example of the racing car in the third column, the IBTM block puts larger weights on the wheel of the car. These observations demonstrate that more informative tokens contribute more to the merged tokens with larger aggregation weights in the token merging process of the IBTM block.

