# OpenReview forum: "Efficient Visual Transformer by Information Bottleneck Inspired Token Merging"
_ICLR.cc/2025/Conference — ICLR 2025 Conference Withdrawn Submission_

### Official Review · Reviewer_b1MG · 2024-10-28

**Soundness:** 3
**Presentation:** 3
**Contribution:** 2
**Rating:** 3
**Confidence:** 5

**Summary:**

This paper proposes an innovative Information Bottleneck inspired Token Merging (IBTM) block for ViTs. IBTM block generates a mask for merging tokens by minimizing the difference between the mutual information of the merged tokens with the original tokens and the corresponding class label. With the IBTM block, the merged tokens can better simulate the behaviour of the original tokens during inference. The proposed approach is adapted to four different pre-trained ViT backbones. Comparisons against three token merging methods demonstrate the effectiveness and generalizability of the IBTM block.

**Strengths:**

1. This paper has compared state-of-the-art token merging methods on various backbones.

2. The proposed token merging strategy with respect to the variation is interesting.

**Weaknesses:**

1.	__Insufficient and unclear motivation__: In the Introduction section, the authors keep arguing that existing token merging methods [1, 2, 3] significantly sacrifice prediction accuracy and claiming their goal is to merge tokens without largely sacrificing performance. However, the reasons why existing token merging methods do not work well and potentially lead to significant performance drops are not clearly explained or justified. As a result, the knowledge gap that IBTM intends to narrow and the drawbacks of existing methods that IBTM is designed to resolve are fairly unclear. In short, the motivation that simply “method A does not work well so we propose a better method B” is weak. The weak motivation essentially affects this manuscript, lowering its academic contribution.

2.	__Imprecise argument__: In the Introduction section, the authors claim that "existing token merging methods largely sacrifice the prediction accuracy of the original transformer networks for reduced computation costs." However, this assertion is misleading, as with an appropriate token merging ratio, these methods do not experience significant accuracy loss. For instance, ToMe [1] achieves a 20% speed-up on ViT-B/AugReg with only a 0.4% accuracy drop, which cannot be described as "largely sacrificing prediction accuracy."

3.	__Disorganized structure and weak readability of the Introduction section__: Following Weaknesses 1 and 2, the whole Introduction section is poorly structured. It is overly repetitive and includes unnecessary content, which detracts from the clarity of the main argument. For instance, Lines 67-96 contain repetitive statements and too many details for the Introduction, which should be summarized for readability. And there is no need for a separate subsection 1.1 on the contributions.

4.	__Unfair comparisons and potential over-claiming__: The authors neglect the finetuning-free nature of some existing token merging methods [1,2] and compare their finetuned IBTM results to the off-the-shelf (i.e., without finetuning) performance of these methods. Such comparisons are not fair and do not accurately reflect the proposed IBTM’s advantages, which also lead to potential over-claiming of the superiority of IBTM.

5.	__Lack of explanation on applying token merging to hierarchical ViTs__: The paper fails to clarify how existing token merging methods, like ToMe [1] and ToFu [2], are adapted for hierarchical ViTs, such as Swin-Transformer [4]. In fact, most token merging (and token pruning) methods are designed for plain-structured ViTs, and applying them to hierarchical models can conflict with window-based self-attention and downsampling layers. In Table 1, where the authors report the performance of these methods on Swin Transformer and other hierarchical ViTs [5], it is necessary to explain how they adapted the existing token merging methods and their proposed IBTM for these hierarchical ViTs.

6.	__Missing technical details__: In the Formulation section, it is unclear whether the token merging mask is dynamically generated w.r.t. different inputs or fixed for all the inputs. If the mask is dynamically generated, based on Equations 3 and 4, I do not think this method would be faster than ToMe. If the mask is fixed after finetuning, it is necessary to analyze the mask pattern, which may reflect some hidden token relationships and bring deeper insights.

7.	__Performance issues__: As shown in Table 7 in Appendix D.2, the proposed IBTM also suffers from a noticeable performance drop when the compression ratio decreases, which undermines its claimed advantages.

8.	__Writing and presentation issues__:

    8.1.	 Inconsistent and confusing terminology: In Lines 15-16, IBTM is introduced as an abbreviation for "Information Bottleneck inspired Token Merging," while in Lines 52-53, it is referred to as "Transformer with Learning Token Merging," causing confusion. In Line 1059, the proposed method is again referred to as LTM.

    8.2.	 Grammar errors: There are quite a few grammatical errors. For instance, in Line 63, the word "less" should be "fewer“. I suggest the authors carefully proofread this manuscript.

    8.3.	Reference formatting errors: In Line 175, the reference format is incorrect at the start of the sentence. In Line 408, ToFu (Kim et al., 2024) is cited twice unnecessarily.

    8.4.	Line 47 cites _Attention Augmented Convolutional Networks_ [6], which was published earlier than ViT and seems irrelevant to the corresponding claim. In addition, some recent token merging-based methods [7,8] should be cited, which provide necessary insights in this direction.

[1] Bolya, Daniel, et al. "Token merging: Your vit but faster." ICLR, 2023.

[2] Kim, Minchul, et al. "Token fusion: Bridging the gap between token pruning and token merging." WACV, 2024.

[3] Bonnaerens, Maxim and Dambre, Joni. "Learned thresholds token merging and pruning for vision transformers." TMLR, 2023.

[4] Liu, Ze, et al. "Swin transformer: Hierarchical vision transformer using shifted windows." ICCV, 2021.

[5] Cai, Han, et al. "Efficientvit: Lightweight multi-scale attention for high-resolution dense prediction." ICCV, 2023.

[6] Bello, Irwan, et al. "Attention augmented convolutional networks." ICCV, 2019.

[7] Xu, Xuwei, et al. "GTP-ViT: Efficient Vision Transformers via Graph-based Token Propagation." WACV, 2024.

[8] Wei, Siyuan, et al. "Joint token pruning and squeezing towards more aggressive compression of vision transformers." CVPR, 2023.

**Questions:**

1. According to Weakness 4, could the author please provide IBTM's performance without finetuning on the four ViT backbones for a fair comparison against ToMe and ToFu?

2. According to Weakness 5, could the author please provide details on adapting existing token merging methods (e.g., ToMe) to hierarchical ViTs (e.g., Swin Transformer), including but not limited to how to perform the downsampling layer, how to split the windows after reduction, how many tokens are there in each layer?

3. According to Weakness 6, could the author please explain how the token merging mask is generated for various inputs?

4. Could the author please compare their model with token pruning methods, such as EViT [1] and ATS [2], under the same experimental environment? EViT and ATS are strong baseline models, which achieve competitive trade-offs between accuracy and efficiency compared to the token merging methods in various standard experiments [3].

5. Could the authors please explain why choosing efficient ViTs (i.e., FLOPs < 1G) as the backbone for experiments rather than large ViTs? Token reduction technologies (including token pruning and token merging) are initially designed to resolve the massive computational complexity problem of ViTs. They should target expediting large ViTs rather than efficient ViTs. I do not find any significant contribution by reducing the FLOPs of EfficientViT from 0.52G to 0.44G after token merging in terms of real-world application.

[1] Liang, Youwei, et al. "Not all patches are what you need: Expediting vision transformers via token reorganizations." ICLR, 2022.

[2] Fayyaz, Mohsen, et al. "Adaptive token sampling for efficient vision transformers." ECCV, 2022.

[3] Haurum, Joakim Bruslund, et al. "Which tokens to use? investigating token reduction in vision transformers." ICCV, 2023.

---

> ### Author Response · Authors · 2024-11-30
> **Response to Reviewer b1MG Part 1**
>
> Thank you for your review
> We appreciate the review and the suggestions in this review. The raised issues are addressed below.
>
> **Responses to the Weaknesses**
>
> **1. “Insufficient and unclear motivation...”**
>
> We respectfully point out that this review missed the important insight and motivation which have been acknowledged by other reviewers such as Reviewer m6sM. We have a very important insight as the motivation for token merging with the IB principle. As evidenced in Table 2 and Table 5 in the paper, existing token merging methods, such as ToMe [1] and LTMP [3], already reduce the IB loss of the base models since they reduce redundant tokens with less relevant information for classification, which adheres to the IB principle that aims at learning representations more correlated with class labels while decreasing their correlation with the inputs. By explicitly reducing the IB loss in the token merging process, IBTM-Transformers enjoy lower IB loss compared to existing token merging methods and show better performance with even lower computational costs.
>
> **2. “Imprecise argument...”**
>
> We respectfully point out that there is a common sense mistake in this review, and there is misunderstanding about the claim “existing token merging methods largely sacrifice the prediction accuracy of the original transformer networks for reduced computation costs.” As a common practice in the token merging and the general model compression literature, we cannot look at the accuracy drop only at a low compression rate, such as a 20% speed-up on ViT-B/AugReg with only a 0.4% accuracy drop mentioned by this reviewer. Instead, we need to examine the performance of the compressed models across various compression rates. Our claim should be understood from a comparative study perspective, which is factually evidenced by the results in Table 1, where IBTM outperforms ToMe with comparable FLOPs/inference time. In addition, Table 8 of the revised paper shows that IBTM, with a 20% speed-up on ViT-B, still outperforms the vanilla ViT-B without compression.
>
> In the revised paper, we clearly indicate that the IBTM causes less drop in prediction accuracy compared to the state-of-the-art token merging methods after this claim. The revised argument reflects this comparative aspect more accurately, acknowledging that while methods like ToMe can achieve efficiency gains with minor accuracy drops, IBTM demonstrates a better trade-off between the computational efficiency and the model performance, benefiting from the IB principle.
>
>  **3. “Disorganized structure and weak readability of the Introduction section...”**
>
> We remark that all the other reviewers appreciate the organization of the instruction section. On the other hand, we respect the opinion of this reviewer and revised the introduction section in the revised paper, focusing on eliminating redundancy and emphasizing key points succinctly. In addition, we integrated the contributions directly into the main text of the Introduction, eliminating the need for a separate subsection.
>
>  **4. “Unfair comparisons and potential over-claiming...”**
>
> We respectfully point out that there is a factual mistake in this review about the unfair comparison and potential over-claiming. As a standard practice in the token merging literature where the token merging models have learnable parameters, these earnable parameters need to be trained in a fine-tuning process. For example, LTMP [3], a token merging method with learnable parameters, loads the pre-trained vision transformers and fine-tunes the additional learnable parameters in the token merging modules. The underlying reason is that these learnable parameters are initialized randomly, so we can only obtain subpar performance without training these parameters. Moreover, it is important to note that the token merging methods without fine-tuning, such as ToMe and ToFu, have a computational step with token matching at all the layers to ensure similar original tokens are merged so that these models still need a computational process before performing token merging.
>
> IBTM requires fine-tuning since it incorporates randomly initialized parameters in the token merging modules that require training. We follow the standard settings of the fine-tuning based token merging in LTMP [3] for our experiments. As evidenced in Table 1, IBTM outperforms the LTMP after fine-tuning for different numbers of epochs. In addition, it is worthwhile to mention that although ToMe [1] and ToFu [2] are free of fine-tuning, they require additional computation costs in performing the token matching at different layers.

---

> > ### Author Response · Authors · 2024-11-30
> > **Response to Reviewer b1MG Part 2**
> >
> > **5. “Lack of explanation on applying token merging to hierarchical ViTs...”**
> >
> > For the experiments with Swin Transformers, the token merging is performed between the multi-head self-attention modules with either regular or shifted windows and the MLP layers. As a result, the input to the MLP layers has fewer tokens, thereby decreasing the computational costs in the MLP layers.  Following the MLP layers, the merged tokens are padded with the same features to restore the original number of tokens. This ensures compatibility with the hierarchical structure of the Swin Transformer, maintaining alignment with subsequent operations such as patch merging and window-based self-attention. This design allows the token merging process to integrate seamlessly with the Swin Transformer architecture while preserving its structural and computational efficiency.
> >
> > **6. “Missing technical details...”**
> >
> > The token merging masks are dynamically generated w.r.t. different inputs following Equation (3) and Equation (4) in Section 3.2. of the paper. As shown in Proposition 3.2, the updating formulation of the token merging mask is based on the features at corresponding layers. Although ToMe is faster than LTMP with the same token compression ratio, models compressed by LTMP exhibit higher classification accuracy than ToMe with an even higher token compression ratio. For example, we set the compression ratio of the LTMP models to $0.7$ and the compression ratio of all the competing token merging methods to $0.75$ for the experiments in Table 1. It is observed that our LTM-Transformers achieve higher top-1 accuracy with even fewer FLOPs and faster inference speed compared to the current state-of-the-art token merging methods.
> >
> > **7. “Performance issues...”**
> >
> > The performance drop with increasing compression is inevitable for token merging. The compression ratio directly affects the number of retained tokens, which has a fundamental impact on the model's representation power. Reducing the number of tokens too aggressively inevitably removes informative content, which leads to a decrease in prediction accuracy. However, the proposed IBTM aims to strike a balance by optimizing the merging process based on the Information Bottleneck principle, ensuring that the merged tokens retain maximum relevant information. This allows IBTM to maintain a competitive performance level, even as we push towards higher compression ratios, outperforming other existing methods in the same context. The aforementioned implementation details have been added to Section E.1 in the appendix of the revised paper.
> >
> > As evidenced in Table 1 in the paper, the IBTM consistently outperforms existing state-of-the-art token merging methods, including ToMe [1], ToFu [2], and LTMP [3]. Although IBTM suffers from performance drops with a higher compression ratio, the performance drop is significantly less severe than existing token merging methods. As shown in Table 7, LTM-ViT-B, with a compression ratio of 0.65, still achieves the same performance as the original ViT-B model.
> >
> > **8. “Writing and presentation issues...”**
> >
> > Thanks for pointing out the errors and typos. We have fixed them in the revised paper. In addition, the two recent works [7, 8] related to token merging are discussed in the related works of the revised paper.
> >
> > **Responses to the Questions**
> >
> > **1. “According to Weakness 4...IBTM's performance without finetuning...”**
> >
> > Please refer to the response to Weakness 4.
> >
> > **2. “According to Weakness 5...details on adapting existing token merging methods (e.g., ToMe) to hierarchical ViTs (e.g., Swin Transformer)...”**
> >
> > Please refer to the response to Weakness 5.
> >
> > **3. ”According to Weakness 6...how is the token merging mask generated for various inputs?”**
> >
> > Please refer to the response to Weakness 6.

---

> > > ### Author Response · Authors · 2024-11-30
> > > **Response to Reviewer b1MG Part 3**
> > >
> > > **4. "...compare their model with token pruning methods, such as EViT [9] and ATS [10]..."**
> > >
> > > We compare LTMP with the token pruning methods EViT [9] and ATS [10] as suggested by the reviewer. For a fair comparison, we apply EViT and ATS on the same pre-trained ViT-B backbone used in Table 1 in the paper, and apply the same data augmentation strategies as reported in Section 4.1. As both EViT and ATS require fine-tuning, we follow the settings in Section 4.1.1 and fine-tune the models compressed with EViT and ATS, that are EViT-ViT-B and ATS-ViT-B for 1, 5, 10, 25, and 50 epochs, respectively, and compare them with IBTM models fine-tuned for the same number of epochs. The results are shown in the table below. It is observed that with an even faster inference speed, the IBTM-ViT-B achieves higher top-1 accuracy on ImageNet compared to the state-of-the-art token pruning methods. For example, IBTM-ViT-B outperforms EViT-ViT-B by $0.37\%$ in top-1 accuracy after fine-tuning for 50 epochs, demonstrating the superiority of the IBTM in compressing vision transformers.
> > >
> > >
> > >
> > > | Methods                     | Inference Time (ms/batch) | Top-1 Accuracy (%) Epoch = 0 | Top-1 Accuracy (%) Epoch = 1 | Top-1 Accuracy (%)  Epoch = 5 | Top-1 Accuracy (%)  Epoch = 10 | Top-1 Accuracy (%)  Epoch = 25 | Top-1 Accuracy (%)  Epoch = 50 |
> > > | --- | --- | --- | --- | --- | --- | --- | --- |
> > > | ViT-B                       | 37.2                      | 83.74                | -                    | -                    | -                     | -                     | -                     |
> > > | EViT-ViT-B                  | 31.9                      | -                    | 82.22                | 82.54                | 83.15                 | 83.49                 | 83.62                 |
> > > | ATS-ViT-B                   | 31.2                      | -                    | 82.49                | 82.85                | 83.05                 | 83.32                 | 83.38                 |
> > > | **IBTM-ViT-B (Fine-tuned)** | **30.7**                      | -                    | **83.35**            | **83.57**            | **83.76**             | **83.91**             | **83.96**             |
> > >
> > > **5. "...explain why choosing efficient ViTs (i.e., FLOPs < 1G) as the backbone for experiments..."**
> > >
> > > In our experiments, we evaluate the effectiveness of IBTM on both large ViTs, such as ViT-B and Swin-B, and efficient ViTs, such as MobileViT and EfficientViT. By applying IBTM on vision transformers of different sizes, we aim to demonstrate the universal applicability and scalability of IBTM across different computational scales and architectural frameworks. The focus on efficient ViTs is particularly relevant for real-world applications that require a balance between accuracy and computational efficiency. These environments often include mobile devices and embedded systems where even modest reductions in computational load  have significant implications. Although efficient vision transformers already exhibit low computational costs, further compressing the models are still critical in environments with limited computational resources, such as mobile and embedded systems.
> > >
> > > **References**
> > >
> > > [1] Bolya, Daniel, et al. "Token merging: Your vit but faster." ICLR, 2023.
> > >
> > > [2] Kim, Minchul, et al. "Token fusion: Bridging the gap between token pruning and token merging." WACV, 2024.
> > >
> > > [3] Bonnaerens, Maxim and Dambre, Joni. "Learned thresholds token merging and pruning for vision transformers." TMLR, 2023.
> > >
> > > [4] Liu, Ze, et al. "Swin transformer: Hierarchical vision transformer using shifted windows." ICCV, 2021.
> > >
> > > [5] Cai, Han, et al. "Efficientvit: Lightweight multi-scale attention for high-resolution dense prediction." ICCV, 2023.
> > >
> > > [6] Bello, Irwan, et al. "Attention augmented convolutional networks." ICCV, 2019.
> > >
> > > [7] Xu, Xuwei, et al. "GTP-ViT: Efficient Vision Transformers via Graph-based Token Propagation." WACV, 2024.
> > >
> > > [8] Wei, Siyuan, et al. "Joint token pruning and squeezing towards more aggressive compression of vision transformers." CVPR, 2023.
> > >
> > > [9] Liang, Youwei, et al. "Not all patches are what you need: Expediting vision transformers via token reorganizations." ICLR, 2022.
> > >
> > > [10] Fayyaz, Mohsen, et al. "Adaptive token sampling for efficient vision transformers." ECCV, 2022.

---

> ### Author Response · Authors · 2024-12-02
> **Reminder of Feedback**
>
> Dear Reviewer b1MG,
>
> This is a gentle reminder of your feedback. All the concerns in your original review have been addressed. Since today is the last day for your feedback, we really look forward to your feedback. We will clarify your further doubts/concerns if there are any. Thank you for your time!
>
> Best Regards,
>
> Authors

---

> > ### Comment · Reviewer_b1MG · 2024-12-03
> >
> > Dear Authors,
> >
> > I have thoroughly reviewed your responses and the revised manuscript. I appreciate the effort put into addressing my concerns, including but not limited to:
> >
> > * Providing experiments on large ViT models.
> >
> > * Providing experiments of ATS and EViT.
> >
> > * Providing explanations to my concerns on the technical details and motivations.
> >
> > * Revising the manuscript.
> >
> > However, I still have some further concerns:
> >
> > __1. Regarding Weakness 1:__
> >
> > My concern was not about the significance of incorporating the Information Bottleneck (IB) principle into token merging methods. Instead, I was pointing out that the primary goal and motivation stated in the Introduction (Lines 054–058) are weak. The stated goal, "to merge the output tokens of all the transformer blocks into fewer tokens without largely sacrificing the prediction accuracy of the original vision transformer," is broadly applicable to all token merging methods and does not highlight the unique value of your work.
> >
> > While you claim strong motivation for token merging with the IB principle, it is essential to explain why existing methods perform poorly in terms of the IB principle. Without this context in the Introduction, the rationale for studying IB is unclear. This critical point is not adequately addressed in either the original or revised manuscript.
> >
> > __2. Regarding Weakness 4:__
> >
> > I fully understand that certain token merging (or pruning) methods require training before being effective, such as DynamicViT. However, my concern was about fairness in your comparisons. Specifically, I was asking for results where ToMe and ToFu are fine-tuned for the same number of epochs as IBTM. Without this, the comparison appears biased. While it is reasonable to claim that IBTM achieves better performance with fewer fine-tuning epochs, claiming superiority with more fine-tuning epochs does not convincingly reflect the advantages of your approach.
> >
> > __3. Regarding Weakness 5:__
> >
> > Your explanation is still unclear to me.
> >
> > 3.1. You mentioned that "merged tokens are padded with the same features to restore the original number of tokens." However, what does the "same features" mean here? Are they the original unmerged tokens?
> >
> > 3.2. If I understand correctly, the padded tokens should not participate in the following computations. But this raises further questions: What happens if all tokens in a window are merged into only a few tokens? How are tokens placed back into the window? How does the shifted window mechanism function in this scenario?
> >
> > For instance, if $N=H\times W$ tokens are merged into $P=H'\times W'$ tokens as mentioned in the manuscript, how are they organized in the hierarchical ViTs? Are there spatial relationships among the $P$ tokens? In the original hierarchical ViTs, the tokens divided into the same window are spatially related.

---

> ### Author Response · Authors · 2024-12-04
> **Further Responses to Reviewer b1MG Part 1**
>
> Thank you for your feedback, and below are our further responses to your remaining concerns.
>
> **(1) Further Response for Weakness 1**
> We will revise the claimed goal, "to merge the output tokens of all the transformer blocks into fewer tokens without largely sacrificing the prediction accuracy of the original vision transformer," to "to merge the output tokens of all the transformer blocks into fewer tokens without largely sacrificing the prediction accuracy of the original vision transformer adhering to a principled Information Bottleneck principle which allows for an informative token merging process". In this way, the unique value of this work is highlighted in the revised goal.
>
> **...why existing methods perform poorly in terms of the IB principle.** The existing token merging methods merge the original tokens into target tokens by the weighted average [1, 2, 3]. Since each target token is an aggregation of the original tokens and the number of target tokens is less than that of the original tokens, the token merging process can be viewed as an information compression process where the rich information in the original tokens is compressed into the limited target tokens. As a result, the target tokens are less correlated with the input training features compared to the original tokens, so the mutual information between the target tokens and the input training features is smaller than that between the original tokens and the input training features, leading to a smaller IB loss compared to the baseline model without token merging. As shown in Table 2 in Section 4.2 and Table 7 in Section E.1 of the appendix, the baseline token merging methods, ToMe and LTMP, can already reduce the IB loss compared to the baseline models.
>
> However, the information compression process is not effective enough in preserving the original rich information in the original tokens in the existing token merging methods such as ToMe and LTMP, because there is no information-theoretic measure to ensure that more informative tokens can receive higher importance weights in the weighted average process for token merging so that the target tokens can retain as much information in the original tokens as possible. This drawback of the existing token merging methods is also quantitatively reflected by the IB loss, that is, the IB loss of the existing token merging methods is not small enough. To this end, we propose IBTM, which employs a novel informative token merging process based on a principled information-theoretic measure, the Information Bottleneck, in the token merging process so that more informative and more important tokens contribute more to the merged tokens with a larger importance weight in the weighted average process for token merging. IBTM quantitatively achieves this goal by further reducing the IB loss by a novel and informative token merging mask which further reduces the IB loss compared to the existing token merging methods.

---

> ### Author Response · Authors · 2024-12-04
> **Further Responses to Reviewer b1MG Part 2**
>
> **(2) Further Response for Weakness 2**
>
> **“I was asking for results where ToMe and ToFu are fine-tuned for the same number of epochs as IBTM.”**
>
> In order to compare ToMe [1] and ToFu [2] with our IBTM in the fine-tuning setup, we
> create two baseline models for ToMe and ToFu. For example, there are two baseline models for ToMe so that these ToMe baseline models can be fine-tuned, which are termed ToMe (backbone only) and ToMe (LTMP mask). ToMe (LTMP mask) employs the learnable mask generated by Sigmoid from the LTMP [3], and the final mask for token merging is the product of the binary mask of ToMe and the learnable mask by LTMP. The weights of the neural backbone and the weights for the sigmoid will be trained when fine-tuning the ToMe (LTMP mask) model. In contrast, ToMe (backbone only) does not use the learnable mask generated from the LTMP [3], and only the weights of the neural backbone will be trained when fine-tuning the ToMe (backbone only) model. Similarly, we have designed two baseline models for ToFu, which are ToFu (backbone only) and ToFu (LTMP mask). We remark that ToMe (LTMP mask) and ToFu (LTMP mask) have the same number of learnable parameters for token merging as our IBTM model when the same neural backbone (such as ViT-B) is used.
>
> We have fine-tuned the two baseline models for both ToMe and ToFu for the same number of epochs as IBTM-ViT-B for 1, 5, 10, 25, and 50 epochs following the settings in Section 4.1.1 of our paper. The results are shown in the table below. It is observed that although fine-tuning can improve the performance of the ToMe and the ToFu models, the IBTM-ViT-B still outperforms the ToMe (backbone only) and ToFu (backbone only) models. For example, IBTM-ViT-B outperforms the ToFu (backbone only) by $0.46$% in the top-1 accuracy when fine-tuned for 50 epochs with even faster inference speed. In addition,  IBTM-ViT-B also outperforms the ToMe (LTMP mask) and the ToFu (LTMP mask). For instance, the IBTM-ViT-B outperforms the ToFu (LTMP mask) by $0.41$% in the top-1 accuracy when fine-tuned for 50 epochs with even faster inference speed, demonstrating the superiority of the IBTM in token merging under the fine-tuning setup.
>
> All the baseline models for both ToMe and ToFu use the same neural backbone (ViT-B) as IBTM-ViT-B.
> | Methods                     | Inference Time (ms/batch) | Top-1 Accuracy (%) Epoch = 0 | Top-1 Accuracy (%) Epoch = 1 | Top-1 Accuracy (%)  Epoch = 5 | Top-1 Accuracy (%)  Epoch = 10 | Top-1 Accuracy (%)  Epoch = 25 | Top-1 Accuracy (%)  Epoch = 50 |
> | --- | --- | --- | --- | --- | --- | --- | --- |
> | ViT-B                       | 37.2                      | 83.74                | -                    | -                    | -                     | -                     | -                     |
> | ToMe (backbone only)                  | 31.0                     | 82.86                | 82.90           | 83.10            | 83.35               | 83.44                 | 83.48               |
> | ToMe (LTMP mask)                 | 31.4                  | -                    | 83.02          | 83.20               | 83.33                | 83.48                 | 83.54              |
> | ToFu (backbone only)                   | 31.5                      | 83.22                | 83.24          | 83.30          | 83.40                | 83.48            | 83.50               |
> | ToFu (LTMP mask)                   | 32.0                | -                    | 83.25          | 82.38               | 83.45               | 83.52               | 83.55              |
> | **IBTM-ViT-B (Fine-tuned)** | **30.7**                      | -                    | **83.35**            | **83.57**            | **83.76**             | **83.91**             | **83.96**             |

---

> ### Author Response · Authors · 2024-12-04
> **Further Responses to Reviewer b1MG Part 3**
>
> **(3) Detailed process in token merging in hierarchical ViTs.**
>
> To answer all your questions, including the raised questions 3.1 and 3.2, we provide a detailed description about the token merging process in hierarchical ViTs as follows.
>
> **The token merging process in hierarchical ViTs.** In the token merging process for hierarchical ViTs, the token merging modules are added between the attention modules and the MLP layers. Let the original tokens $X \in R^{N\times D}$ be the features before token merging, where $N=H\times W$ is the number of tokens. Let $\tilde X \in R^{P\times D}$ be the features after the token merging, where $P$ is the number of merged tokens. **We explain how to obtain the output features of size $N \times D$ from $\tilde X$ as follows.**
>
> $\tilde X$ will be fed into the MLP layers in the transformer block. Let $\tilde Z = MLP(\tilde X)\in R^{P\times D}$ be the output features of the MLP layers. Before feeding $\tilde Z$ to the window attention of the next layer, we need to pad the features $\tilde Z$ so that the spatial size of $\tilde Z$ is still $N = H \times W$. Let $Z$ denote the padded version of $\tilde Z$. This padding process is achieved by repeating the merged tokens at the positions where the original tokens are merged into them. For example, if two original tokens, $X_j$ and $X_k$, are merged into a target token $\tilde X_i$ in the token merging process, both $Z_j$ and $Z_k$ co-located at the positions of $X_j$ and $X_k$ in the feature map will be filled by copying the target token $\tilde Z_i$. In this way, the features $\tilde Z$ are padded to the feature map $Z$ with a shape of $N \times D$ again, which can be reshaped to $H\times W \times D$. Therefore, the tokens in the padded features $Z$ as the input to the window attention in the next transformer block have the same spatial relationship as the original tokens in $X$. In addition, the computational cost is reduced since the number of tokens processed in the MLP layers is reduced from $N$ to $P$.
>
> To answer your question 3.1, the “same features” are the copied target tokens in the description titled **The token merging process in hierarchical ViTs**.
>
> To answer your question 3.2, please refer to the token merging process in the same description above. Even though the tokens in a window are merged into only a few tokens, the input features to the next window attention will still be in the same shape as the vanilla swin model due to the padding strategy. Therefore, the features $Z\in R^{N\times D}$ are of the same shape as the vanilla swin model, and the tokens can be placed into the window in the same manner as in the vanilla swin model. Again, as mentioned in **The token merging process in hierarchical ViTs**, the tokens in the padded features $Z$ as the input to the window attention in the next transformer block have the same spatial relationship as the original tokens as the input to the current transformer block.
>
> **References**
>
> [1] Bolya, Daniel, et al. "Token merging: Your vit but faster." ICLR, 2023.
>
> [2] Kim, Minchul, et al. "Token fusion: Bridging the gap between token pruning and token merging." WACV, 2024.
>
> [3] Bonnaerens, Maxim and Dambre, Joni. "Learned thresholds token merging and pruning for vision transformers." TMLR, 2023.

---

### Official Review · Reviewer_1etX · 2024-11-02

**Soundness:** 3
**Presentation:** 3
**Contribution:** 3
**Rating:** 8
**Confidence:** 5

**Summary:**

The authors propose a novel and compact transformer block (IBTM) that implements token merging in a learnable manner. IBTM is compatible with various popular and compact transformer architectures, effectively reducing both FLOPs and inference time in vision transformers while maintaining or even enhancing prediction accuracy.

**Strengths:**

1. The authors' motivation is intuitive, and their theoretical analysis from the perspective of the Information Bottleneck framework adds solid support to their claims.

2. The experimental are comprehensive (in particularly, there are detection and segmentation in the appendix) and the results indicate that the proposed approach achieves notable gains and advantages.

3. The authors provide corresponding open-source code, enhancing the credibility and reproducibility of their results.

**Weaknesses:**

1. The related work on *efficient vision transformers* could be more comprehensive, I suggest including relevant studies such as [1].

2. The description of the method could be more concise and clearer; for example, Figure 1 is a bit too brief in its presentation.


[1] Wu X, Zeng F, Wang X, et al. PPT: Token Pruning and Pooling for Efficient Vision Transformers. arXiv preprint arXiv:2310.01812, 2023.

**Questions:**

I wonder if incorporating token pruning techniques could lead to better results?

---

> ### Author Response · Authors · 2024-11-30
> **Response to Reviewer 1etX**
>
> We appreciate the review and the suggestions in this review. The raised issues are addressed below.
>
> **Responses to the Weaknesses**
>
> **1. “The related work on efficient vision transformers could be more comprehensive...”**
>
> In Section 2.1 of the revised paper, we have conducted more comprehensive discussions on the related works on efficient vision transformers, including [1].
>
> **2. “The description of the method could be more concise and clearer...”**
>
> We appreciate the suggestion, and we promise to revise the method section to be more concise and clearer in the final revision.
>
> **Responses to the Questions**
>
> **1. “I wonder if incorporating token pruning techniques could lead to better results?”**
>
> Token merging methods can be easily combined with token pruning methods. For instance, token pruning modules can be applied right after token merging modules in transformers to further compress the vision transformers. We have performed a comparison between the token pruning methods and the IBTM in Section B.4 of the appendix of the revised paper. We will further investigate the effectiveness of combining the token merging methods with the IBTM.
>
> **References**
>
> [1] Wu X, Zeng F, Wang X, et al. PPT: Token Pruning and Pooling for Efficient Vision Transformers. arXiv preprint arXiv:2310.01812, 2023.

---

> > ### Comment · Reviewer_1etX · 2024-12-03
> > **Thank for the Response**
> >
> > Dear authors,
> >
> > Tnanks for your response, I tend to keep my score.
> >
> > Good luck!
> >
> > Reviewer

---

### Official Review · Reviewer_M7pa · 2024-11-03

**Soundness:** 3
**Presentation:** 3
**Contribution:** 2
**Rating:** 6
**Confidence:** 5

**Summary:**

This paper introduces a new token merging method on vision transformer, which leverages information bottleneck theory to learn the merging mask. A novel upper bound of IB is derived to optimize the mask module. Experiments of various ViT variants are conducted on ImageNet to validate the effectiveness.

**Strengths:**

1. This paper proposes a learnable way to merge tokens, which is inspired by information bottleneck and is theoretically feasible for achieving better compression-performance balance compared to previous training-free methods.

2. The performance improvements are consistent and significant.

**Weaknesses:**

1. The major concern of this method is the requirement of training. According to Table 1, the performance does not get satisfied even after fine-tuning for 50 epochs, this raises concern for the computation cost.

2. Lack of experiments to validate the transferrability. ViTs, as the foundation models, are widely adopted in downstream tasks such as transfer learning, object detection and visual-language understanding. As the token merging is trained on ImageNet, it is unknown whether this preference is generic to other tasks and whether the method is better than other token merging methods on transfer tasks.

**Questions:**

3. What is the superiority of the proposed upper bound of IB objective compared to the existing upper bounds such as CLUB [1]?


[1] Cheng, Pengyu, et al. "Club: A contrastive log-ratio upper bound of mutual information." International conference on machine learning. PMLR, 2020.

---

> ### Author Response · Authors · 2024-11-30
> **Response to Reviewer M7pa**
>
> We appreciate the review and the suggestions in this review. The raised issues are addressed below.
>
> **Responses to the Weaknesses**
>
> **1. “The major concern of this method is the requirement of training...”**
>
> As a standard practice in the token merging literature where the token merging models have learnable parameters, these learnable parameters need to be trained in a fine-tuning process. For example, LTMP [1], a token merging method with learnable parameters, loads the pre-trained vision transformers and fine-tunes the additional learnable parameters in the token merging modules. The underlying reason is that these learnable parameters are initialized randomly, so we can only obtain subpar performance without training these parameters. Moreover, it is important to note that the token merging methods without fine-tuning, such as ToMe and ToFu, have a computational step with token matching at all the layers to ensure similar original tokens are merged so that these models still need a computational process before performing token merging.
>
> **2. “Lack of experiments to validate the transferrability...”**
>
> We respectfully point out that we have adopted the IBTM models pre-trained on ImageNet as the feature backbone for other tasks. Our work focuses on performing token merging in vision transformers for computer vision tasks, and we have assessed the transferability of IBTM models on object detection and semantic segmentation in Section A.2 and Section A.3 of the appendix of our paper. We adopt the IBTM models pre-trained on ImageNet as the feature backbone for the object detection and semantic segmentation networks. It is observed in Table 4 and Table 5 that the IBTM models show consistent improvements over different vision tasks.
>
> **Responses to the Questions**
>
> **3. “...compared to the existing upper bounds such as CLUB [1]?”**
>
> Although CLUB [2] also proposes an upper bound for the Information Bottleneck (IB), the derivation of the upper bound in CLUB assumes that the $p(\tilde X|X)$ is known, where $\tilde X$ and $X$ are the random variables representing the learned feature and the input feature. When $p(\tilde X|X)$ is unknown, they adopt the Gaussian Mixture Model (GMM) parameterized by an external neural network to approximate the upper bound. In contrast, the variational upper bound for the IB derived in our paper does not have such an assumption. As shown in Lemma C.1 in Section C.2 of the appendix of our paper, $p(\tilde X|X)$ can be directly computed from the training data without the need for training another neural network.
>
> **References**
>
> [1] Bonnaerens, Maxim and Dambre, Joni. "Learned thresholds token merging and pruning for vision transformers." TMLR, 2023.
>
> [2] Cheng, Pengyu, et al. "Club: A contrastive log-ratio upper bound of mutual information." International conference on machine learning. PMLR, 2020.

---

> > ### Comment · Reviewer_M7pa · 2024-12-03
> > **Thanks for the Response**
> >
> > Dear Authors,
> >
> > Thank you for your detailed response. Most of my concerns are addressed. I have carefully read the comments and responses of other reviewers, and decided to raise my confidence to 5 but keep my rating.
> >
> > Reason for not giving a higher score:  The method clearly has better performance than previous method LTMP, which also requires fine-tuning, so I leaned towards acceptance of this paper considering the performance strength and technical novelty. However, the tine-tuning requires additional training, which poses concerns on the scalability and generalizability, so I decided not giving a higher score.
> >
> > Thanks,
> > Reviewer

---

### Official Review · Reviewer_m6sM · 2024-11-03

**Soundness:** 3
**Presentation:** 2
**Contribution:** 3
**Rating:** 6
**Confidence:** 4

**Summary:**

This paper studied the problem of token merging/pruning in vision transformers. The authors took inspiration from the connection between token merging and information bottleneck (IB) and introduced a new strategy to leverage the IB principle to guide the token merging process. They derived a variational upper bound loss from the IB loss and proposed optimizing this objective to generate token merging masks. Experiments on ImageNet dataset with multiple tiny-/small-size vision transformers showed considerable improvement of the proposed method over prior works.

**Strengths:**

1. The paper is well-motivated. Connecting token merging with the IB principle is straightforward and makes a lot of sense. An algorithm guided by this principle is novel and might bring new insights into this research direction.

2. The derivation of the variational upper bound for the IB loss looks technically sound. Empirical analysis also suggests that reducing this upper bound indeed results in lower IB losses for token merging.

3. Experiments on ImageNet dataset with multiple tiny-/small-size vision transformers, and a few experiments on MS-COCO object detection/instance segmentation, showed considerable and consistent improvements in the proposed method over prior works, sometimes by a ;large margin.

**Weaknesses:**

1. The proposed method, IBTM, requires either additional finetuning epochs to repurpose pretrained models for token merging or extra training time (~30%) to train the vision transformers and the token merging module from scratch.

2. IBTM uses hard masks for token merging, which discard a certain amount of the tokens during inference. I wonder whether soft masks will also work for the same purpose. Additionally, the models with IBTM (thus fewer tokens) perform even better than the original models with more tokens, which is somewhat counter-intuitive. Are there any explanations for these behaviors?

**Questions:**

1. Table 1 suggests that using IBTM will not increase the number of parameters compared to the original models yet Tables 3 and 4 report different scenarios. What is the cause of this difference?

2. Why IBTM is faster than LTMP in inference? These two methods should have the same parametrization scheme and inference pipeline from my understanding of this work.

---

> ### Author Response · Authors · 2024-11-30
> **Response to Reviewer m6sM Part 1**
>
> We appreciate the review and the suggestions in this review. The raised issues are addressed below.
>
> **Responses to the Weaknesses**
>
> **1. “...additional finetuning epochs...extra training time  to train from scratch...”**
>
> As a standard practice in the token merging literature where the token merging models have learnable parameters, these learnable parameters need to be trained in a fine-tuning process. For example, LTMP [3], a token merging method with learnable parameters, loads the pre-trained vision transformers and fine-tunes the additional learnable parameters in the token merging modules. The underlying reason is that these learnable parameters are initialized randomly, so we can only obtain subpar performance without training these parameters. Moreover, it is important to note that the token merging methods without fine-tuning, such as ToMe and ToFu, have a computational step with token matching at all the layers to ensure similar original tokens are merged so that these models still need a computational process before performing token merging.
>
> **2. “...whether soft masks will also work for the same purpose...IBTM (thus fewer tokens) perform even better than the original models...”**
>
> IBTM first generates a token binarized mask $M^{(l)}$ for the $l$-th IBTM block. This mask is then elementwise multiplied by $\tilde G^{(l)}$, which is derived from the gradient on the variational upper bound of the IB loss. Since $\tilde G^{l}$ is a softmask, the token merging mask used by the IBTM block is indeed a softmask.
> In addition, it has been observed in the model compression literature [1,2] that compressing parameter-heavy models, such as vision transformers, can even improve the accuracy rather than compromising it due to the reduced over-fitting effect in the compressed models. In addition, features learned by the IBTM models exhibit lower IB loss compared to the vanilla models, as evidenced in Table 2 of the paper. Lower IB loss indicates a more effective representation of information, which could further explain the superior performance of the IBTM models.
>
> **Responses to the Questions**
>
> **1. “Table 1 suggests that using IBTM will not increase the number of parameters compared to the original models yet Table 3 and 4 report different scenarios... ”**
>
> IBTM first generates a token binarized mask $M^{(l)}$ for the $l$-th IBTM block, employing the same module as used in the LTMP [3]. This mask is then multiplied by $\tilde G^{(l)}$, which is derived from the gradient on the variational upper bound of the IB loss. Therefore, IBTM incorporates the same number of additional parameters as LTMP. Nevertheless, the increase in parameters is marginal and does not alter the first digit after the decimal point of the number of parameters. The parameter sizes reported in Tables 3 and 4 were incorrect due to typographical errors and they have been corrected in the revised version of the paper.
>
> **2.“Why IBTM is faster than LTMP in inference...“*
>
> IBTM demonstrates faster inference compared to LTMP because we configure the token compression ratio of IBTM to be lower than that of LTMP to assess the performance of IBTM with lower FLOPs/ inference time. For our experiments in Table 1, we set the compression ratio for IBTM models at $0.7$ and for all other competing token merging methods at $0.75$. It is observed that our IBTM models not only achieve superior top-1 accuracy but also do so with fewer FLOPs and enhanced inference speed, outperforming state-of-the-art token merging methods.
>
> **References**
>
> [1] Chen, Tianlong, et al. "Chasing sparsity in vision transformers: An end-to-end exploration." Advances in Neural Information Processing Systems 34 (2021): 19974-19988.
>
> [2] Kong, Zhenglun, et al. "Spvit: Enabling faster vision transformers via latency-aware soft token pruning." European conference on computer vision. Cham: Springer Nature Switzerland, 2022.
>
> [3] Bonnaerens, Maxim and Dambre, Joni. "Learned thresholds token merging and pruning for vision transformers." TMLR, 2023.

---

> > ### Comment · Reviewer_m6sM · 2024-12-03
> >
> > Thank you for your response. After carefully reading the comments from other reviewers and the authors' responses to all reviewers, I think that my raised concerns are resolved, and remain positive about the acceptance of this paper. Overall, this work has a clear motivation and presents an effective method for token merging. While bearing extra training time and the methodology itself is not particularly novel/exciting, it shows decent results across multiple benchmarks compared to all the baselines.
> >
> > Thus, I decided to maintain my original score.

---

### Comment · Area_Chair_k2Jk · 2024-12-03
**Discussion due soon**

Dear all reviewers,

Our reviewer-author discussion will end soon. For each of you, please check all the files and see if anything you'd like to discuss with authors.

Best,
Your AC

---

### Author Response · Authors · 2024-12-04
**Summary of Revisions**

Dear AC and Reviewers,

Thank you for your time reviewing and handling this paper. We would like to let you know that we have addressed all the concerns in the reviews.

Regarding the following remaining concerns from Reviewer b1MG,

(1) highlighting the value of this work in the statement of goal, and explain why existing methods perform poorly in terms of the IB principle;

(2) comparing the fine-tuned ToMe and ToFu with our IBTM;

(3) detailed description about the token merging process by IBTM in hierarchical ViTs,

we have provided **Further Responses to Reviewer b1MG Part 1-Part 3** addressing the above three concerns, respectively,
in our response to Reviewer b1MG. We sincerely hope that Reviewer b1MG and you could review such further responses.

Thank you again for your time!

Best Regards,

The Authors

---

### Note · Authors · 2025-01-22

I have read and agree with the venue's withdrawal policy on behalf of myself and my co-authors.